# A study on phonemes recognition method for Mandarin pronunciation based on improved Zipformer-RNN-T(Pruned) modeling

**Zhaohui Du**[1], **Xiaofeng Zhao**[1], **Lin Li**[1], **Baohua Yu**[1]*, **Lijiang Miao**[2]*

**1** School of Information Science and Technology, Shihezi University, Shihezi, China, **2** Xinjiang Uygur Autonomous Region Education Examination Centre, Urumqi, China

* ybh_inf@shzu.edu.cn (BY); miaolijiang00@163.com (LM)

**Data availability statement:** The data used in this study is publicly available on the AISHELL platform. The dataset includes. 1. AISHELL1

## Abstract

In recent years, empowered by artificial intelligence technologies, computer-assisted language learning systems have gradually become a hot topic of research. Currently, the mainstream pronunciation assessment models rely on advanced speech recognition technology, converting speech into phoneme sequences, and then determining mispronounced phonemes through sequence comparison. To optimize the phoneme recognition task in pronunciation evaluation, this paper proposes a Chinese pronunciation phoneme recognition model based on the improved Zipformer-RNN-T(Pruned) architecture, aiming to improve recognition accuracy and reduce parameter count. First, the AISHELL1-PHONEME and ST-CMDS-PHONEME datasets for Mandarin phoneme recognition through data preprocessing. Then, three layers of the Zipformer Block architecture are introduced into the Zipformer encoder to significantly enhance model performance. In the stateless Pred Network, the GELU activation function is adopted to effectively prevent neuron deactivation. Furthermore, a hybrid Pruned RNN-T/CTC Loss fusion strategy is proposed, further optimizing recognition performance. The experimental results demonstrate that the method performs excellently in the phoneme recognition task, achieving a Word Error Rate (WER) of 1.92% (Dev) and 2.12% (Test) on the AISHELL1-PHONEME dataset, and 4.28% (Dev) and 4.51% (Test) on the ST-CMDS-PHONEME dataset. Moreover, the model requires only 61.1M parameters, striking a balance between performance and efficiency.

## 1 Introduction

With the increasing maturity of speech recognition technology and the growing popularity of the "Chinese language fever," how to leverage speech recognition technology to more effectively assess learners' Chinese pronunciation proficiency has become a research hotspot in the field of education. Traditional assessment methods rely on manual evaluation, which is not only time-consuming and labor-intensive but also lacks a consistent and unified evaluation standard. To overcome these limitations, researchers have applied pronunciation assessment

dataset retrieval address: https://www.aishelltech.com/kysjcp. 2. MUSAN dataset retrieval address: https://www.openslr.org/17/. The dataset includes various types of speech used for model training, validation, and evaluation.

**Funding:** This research was funded by [the Bing-tuan Science and Technology Public Relations Project "A Data-driven Regional Smart Education Service Key Technology Research and Application Demonstration"] grantnumber [2021AB023]. The funders had no role in study design, data collection and analysis, decision to publish, or preparation of the manuscript.

**Competing interests:** The authors have declared that no competing interests exist.

technology to recognize learners' pronunciation, extracting key features from the speech data. Based on these key features, an analysis of the learners' pronunciation is conducted, identifying specific errors in their speech. Currently, pronunciation assessment methods can be classified into three categories: feature-based pronunciation assessment, GOP (Graphical Output Probability)-based pronunciation assessment, and speech recognition-based pronunciation assessment [1,2]. Among these, speech recognition-based pronunciation assessment methods are characterized by low resource consumption, low latency, and high performance, making them the mainstream evaluation approach. The core of this method lies in utilizing advanced speech recognition technology to construct phoneme recognition models. These models can convert learners' pronunciations into phoneme sequences, thereby enabling precise analysis of pronunciation details. Phoneme recognition, as a prerequisite task for mainstream pronunciation assessment methods, plays a significant role in advancing the development of the pronunciation evaluation field and improving learners' pronunciation skills [3].

Phoneme recognition is the process of identifying phoneme units using advanced speech recognition technology. In recent years, the significant advancements in speech recognition technology have been largely driven by the development of deep learning. With the introduction of deep neural networks (DNN) [4] and convolutional neural networks (CNN) [5], the ability of speech recognition systems to model speech time-series information has been significantly enhanced. In addition, the introduction of end-to-end models such as CTC [6], RNN-T [7], and LAS [8] has further simplified the complexity of the models, enabling speech recognition systems to exhibit stronger performance when processing long sequences. With the advancement of speech recognition technology, phoneme recognition technology has also been correspondingly improved. Researchers have suggested various deep learning-based phoneme recognition models, such as temporal modeling methods based on recurrent neural networks (RNN) [9], and self-attention mechanisms based on Transformers [10]. Despite the significant advancements made by the aforementioned methods in the field of speech recognition, their adaptability to Chinese phoneme recognition tasks remains inadequate. Existing mainstream models often have large parameter counts and complex structures, making them difficult to deploy in practical environments that require high resource efficiency and quick response times, such as language learning applications, online evaluation platforms, and mobile devices. Therefore, there is an urgent need to develop a lightweight model that combines strong modeling capabilities with a low real-time factor, which is crucial for the advancement of Chinese phoneme recognition systems.

Currently, academic research primarily focuses on phoneme recognition for English and other Western languages, while research on Chinese phoneme recognition is relatively limited. This situation has resulted in a scarcity of high-quality datasets specifically for Chinese phoneme recognition. Compared to other languages, the phonetic system of Chinese is significantly more complex. Firstly, Chinese contains a large number of consonants and vowels that are phonetically similar, which makes them difficult to distinguish and prone to confusion. Secondly, tones play a crucial role in the semantic structure of Chinese, with the same syllable having completely different meanings depending on the tone [11]. For example, "ma" represents "mother" "hemp" "horse", and "scold" in the first to fourth tones, respectively. This reliance on tones to distinguish meanings greatly increases the difficulty of fine-grained modeling in Chinese phoneme recognition systems. Meanwhile, in practical application scenarios such as language learning and speech evaluation platforms, Chinese phoneme recognition systems also face the demand for strong real-time feedback. Learners typically expect immediate feedback on their pronunciation as soon as they finish speaking, so they can promptly adjust their pronunciation. This human-computer interaction scenario imposes a very low

real-time factor (RTF) requirement on the system to ensure a smooth and natural user experience. Therefore, Chinese phoneme recognition tasks face two main challenges: first, the high requirement for modeling accuracy due to subtle phonetic differences such as tones; second, the strict limitations on inference speed and response delay in application scenarios. This means that phoneme recognition models must not only have the ability to model complex phonetic features accurately but also complete fast inference and output within a limited time budget to maintain efficient interactivity and practicality.

To address these challenges, this paper adopts the Zipformer architecture, which has shown outstanding performance in the field of speech recognition in recent years. Compared to the traditional Transformer, Zipformer significantly reduces computational complexity while maintaining its ability to model temporal information through a time-compression mechanism. In contrast to RNN-based models, it demonstrates higher efficiency in handling long-range dependencies and parallel computation. Additionally, Zipformer's multi-layer compression and decompression structure can accommodate the modeling needs of multi-dimensional information such as consonants, vowels, and tones in Chinese pronunciation, helping to capture subtle phonetic differences and trends. Moreover, Zipformer's modular structure and excellent scalability make it easier to implement fast inference in online pronunciation evaluation systems, meeting the requirements for real-time feedback. Therefore, in the task of Chinese phoneme recognition, Zipformer strikes a superior balance in terms of accuracy, latency, and model size compared to other mainstream models, making it the preferred model framework for this research.

Based on the aforementioned issues and challenges, the contributions of this paper are as follows:

- In view of the scarcity of Chinese phoneme recognition data resources, this paper is based on the Mandarin speech dataset AISHELL-1 and ST-CMDS. It utilizes the mapping relationship between Chinese characters and phonemes to replace the original character labels with phoneme labels. To address the pronunciation variations of polyphonic characters in different phrases or contexts, a phrase-level phoneme mapping table is used for label substitution, rather than character-by-character replacement. This results in the construction of the AISHELL1-PHONEME dataset and the ST-CMDS-PHONEME dataset, which is labeled with Chinese phonemes and provides data support for subsequent phoneme recognition model training

- In terms of the encoder, this paper proposes a deep Zipformer Block module based on the Zipformer model. Through experimental analysis of the original Zipformer paper, it was found that the number of Encoder Blocks is positively correlated with both the model's parameter size and recognition accuracy. That is, increasing the number of Encoder Blocks improves recognition accuracy but also increases the number of parameters. To address this, this paper designs a three-layer Zipformer Block structure, aiming to maintain recognition accuracy while reducing the number of parameters.

- In terms of the decoder, this paper proposes a GELU-based Pred Network module. To address the issue where the ReLU activation function in the stateless Pred Network may lead to neuron deactivation, this paper introduces the GELU activation function, which has demonstrated excellent performance in natural language processing. Unlike ReLU, GELU smooths and filters the input values through a Gaussian distribution, avoiding the drawback of ReLU's direct truncation of negative values. This enhances sensitivity to input variations and improves the network's expressive capacity and performance stability.

- In terms of loss calculation, this paper proposes a hybrid Pruned RNN-T/CTC loss function. Based on the Pruned RNN-T loss function (including Simple Loss and Pruned Loss),

the CTC loss function is introduced to enhance the model's adaptability to long sequences and variable-length inputs. By experimentally adjusting the weighting ratio, the Pruned RNN-T loss and the CTC loss are combined through weighted fusion to optimize the model's recognition accuracy.

The structure of this paper is as follows: Sect 2 primarily introduces the related applications and research on phoneme recognition technology. Sect 3 presents the methods proposed in this paper. Sect 4 includes comparative experiments, simulated noise experiments, ablation studies, inference experiments, and Chinese phoneme analysis. Finally, Sect 5 concludes the paper.

## 2 Related works

Phoneme recognition technology is becoming increasingly important in pronunciation error detection. It relies on highly accurate speech recognition technology to achieve efficient conversion from speech to phonemes, providing a foundation for speech assessment and language teaching. This section will explore the research and applications of phoneme recognition, detailing the progress of English phoneme recognition technology, and discussing Mandarin phoneme recognition technology along with confusion analysis in practical applications.

### 2.1 Applications of phoneme recognition

Phoneme-level pronunciation accuracy assessment methods are widely applied in the field of pronunciation evaluation. Leung et al. proposed an end-to-end phoneme recognition system based on CNN-RNN-CTC, combining convolutional neural networks (CNN), recurrent neural networks (RNN), and connectionist temporal classification (CTC) technology [12], for detecting and diagnosing mispronunciations in second language learners. Building on this, Zhang et al. proposed an end-to-end automatic pronunciation error detection system based on an improved hybrid CTC/Attention architecture by combining the advantages of CTC (Connectionist Temporal Classification) and attention mechanisms [13]. This system effectively avoids the complex module segmentation and forced alignment requirements commonly found in traditional speech recognition systems. Meanwhile, Yan et al. focused on pronunciation error detection for second language (L2) English learners and proposed an end-to-end approach based on a novel anti-phone modeling technique [14]. Traditional ASR methods often struggle to accurately recognize approximate or distorted pronunciations between native language (L1) and target language (L2) phonemes. The system proposed in this paper enhances the system's comprehensive recognition ability for both categorical and non-categorical mispronunciations by expanding the original L2 phoneme set to include the corresponding anti-phone set.

### 2.2 English phoneme recognition

Speech recognition-based English phoneme recognition technology has developed rapidly. With the rise of deep learning techniques, convolutional neural networks (CNN) have been introduced into phoneme recognition. Abdel-Hamid et al. proposed a CNN-based acoustic model [15], which significantly enhanced the model's adaptability to complex acoustic environments. CNNs excel in phoneme recognition tasks by automatically extracting local features to capture important patterns in audio signals. In addition, Tóth et al. used a hierarchical convolutional deep Maxout network [16], which further enhanced the model's ability

to recognize phonemes. Building upon convolutional networks, Passricha et al. introduced Convolutional Support Vector Machines (CSVM) [17], attempting to combine the benefits of convolutional feature extraction and support vector machines to improve phoneme recognition performance. CSVM integrates the feature extraction capabilities of convolutional layers with the classification power of SVM, providing a new approach to phoneme recognition. Ravanelli et al. proposed SincNet [18], an interpretable convolutional filter model that better captures frequency information in audio signals. SincNet improves frequency feature modeling by using the Sinc function as a filter in the convolutional layers. In the integration of convolutional and recurrent networks, Zhao et al. proposed a Recurrent Convolutional Neural Network (RCNN) [19], which leverages the advantages of both convolutional and recurrent layers in processing speech signals, thereby improving phoneme recognition accuracy.

## 2.3 Chinese phoneme recognition

Mandarin phoneme recognition and confusion analysis provide targeted teaching guidance for Chinese language teachers. Ding et al. collected learners' self-reported pronunciation difficulties and coping strategies through methods such as questionnaires, reading tasks, and recall interviews [20]. The study found that learners generally face challenges in consonants, vowels, and tones during pronunciation, but there is a discrepancy between self-reports and actual performance. Jiang et al. explored the correlation between the pronunciation of Mandarin consonants and the distance in speech signals, revealing the causes of consonant pronunciation confusion from the perspective of signal processing [21]. The experimental results show that confusable consonant pairs (such as /l/ and /n/) have the shortest signal distance at the signal level, which aligns with the fact that they are prone to confusion during pronunciation. Arkin et al. conducted an in-depth analysis of the phonemes in the pronunciation of 50 Uyghur native speakers learning Mandarin Chinese using automatic speech recognition technology [22]. The study found that Uyghur students often misidentify the target phoneme "a" as "e" or "i" when speaking Mandarin, and misidentify "i" as "e" or "ie".

## 3 The proposed model

Given the complexity of Chinese pronunciation and the need for rapid feedback in phoneme recognition, and considering that the phoneme recognition model needs to be deployed on resource-constrained devices, this paper proposes an end-to-end Chinese phoneme recognition model based on the improved Zipformer-RNN-T(Pruned). The model combines the efficient Zipformer encoder with a pruned and optimized RNN-T decoder. Compared to traditional models, this model has advantages in terms of recognition accuracy, model parameters, and computational speed. The overall structure of the model is shown in Fig 1, which is primarily divided into two parts: the encoder and the decoder. First, after feature extraction, the generated FBank features are input into the Zipformer encoder. The first stage of the encoder uses the Conv2-Embed module to downsample the input 100 Hz audio signal to 50 Hz. Then, the Zipformer-Blocks module models the features at a frame rate of 50 Hz. Subsequently, the features undergo further processing through five Encoder-Block module groups, each of which sequentially performs downsampling, Zipformer-Blocks, upsampling, and Bypass modules, with frame rates of 25 Hz, 12.5 Hz, 6.25 Hz, 12.5 Hz, and 25 Hz, respectively. Finally, the features are downsampled to 25 Hz for output. In the decoder section, the input consists of the features processed by the encoder (am) and the phoneme labels. First, the phoneme labels are processed through the Pred Network, producing label features (lm). Then, am and lm are input into the Trivial Joiner module, which calculates the Simple Loss. Based on am, lm, and Simple Loss, the pruning boundary is computed to obtain Pruned Am and

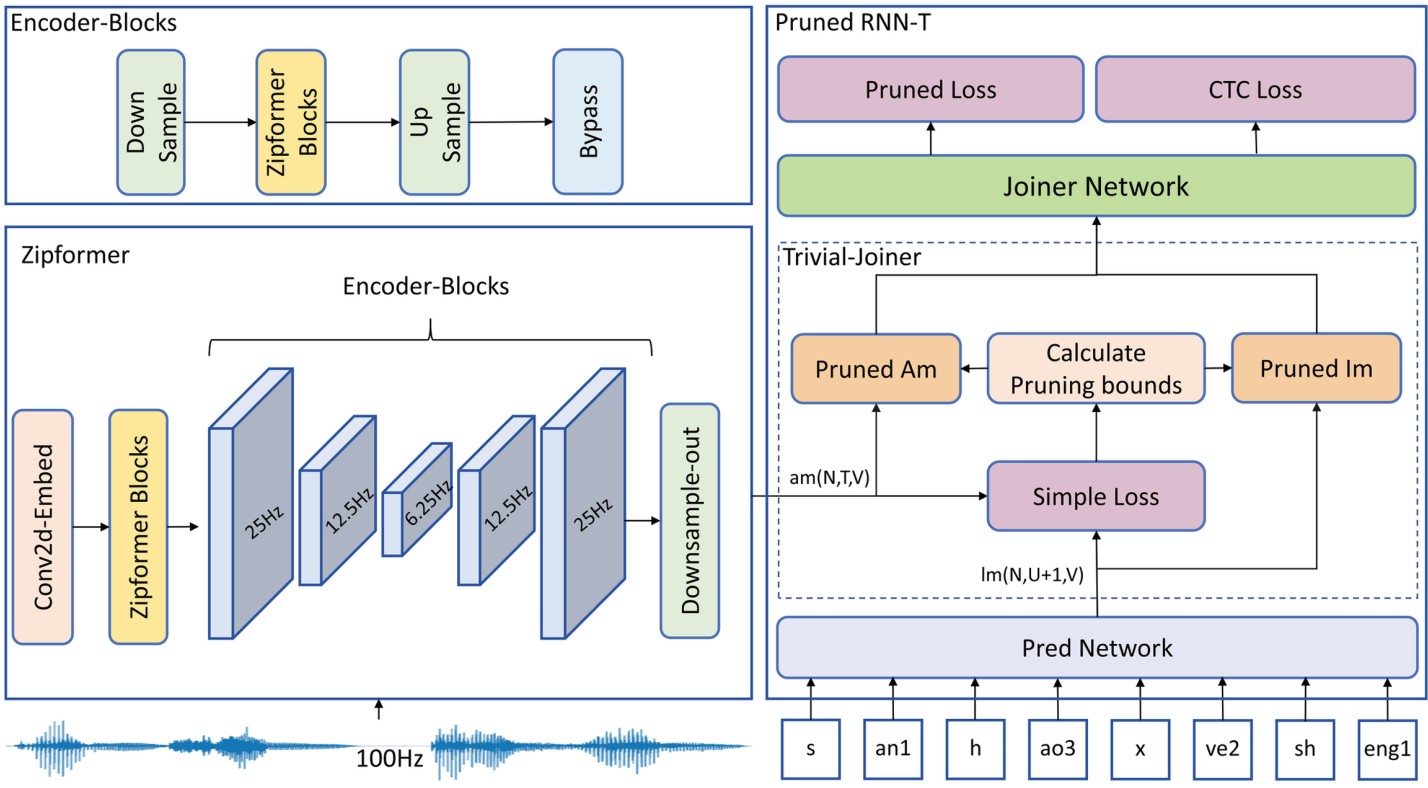

**Fig 1. The overall structure of the Zipformer-RNN-T(Pruned) model.**

Pruned Lm. Finally, Pruned Am and Pruned Lm are input into the Joiner Network for fusion, and the Pruned Loss and CTC Loss are calculated separately. To enhance the model's performance, this paper proposes improvements to the Zipformer Block and Pred Network, and introduces a hybrid Pruned RNN-T/CTC Loss strategy that integrates Simple Loss, Pruned Loss, and CTC Loss to optimize the end-to-end training process. Through this enhancement, the model is able to effectively learn the mapping relationship between audio features and phoneme labels, thereby achieving efficient Chinese phoneme recognition.

### 3.1 Deep Zipformer block

The Zipformer encoder models features across different speech frames, which results in a relatively large number of parameters while maintaining performance. To address this issue, this paper optimizes the Zipformer model by reducing the number of Encoder Blocks to decrease the overall model size. Additionally, a strategy is introduced that reuses the attention weights calculated by MHAW (Multi-Head Attention Weight) to save computational resources. The Zipformer Block's two-layer SCF (Self-Attention, Convolution, and Feed-forward) module group is expanded to three layers, aiming to improve the overall performance of the model by increasing its depth.

The improved Zipformer Block is shown in Fig 2. The speech features are input into the MHAW module to compute the attention weights, which are then passed into the Self Attention module and the NLA (Non-Linear Attention) module within the SCF module group.

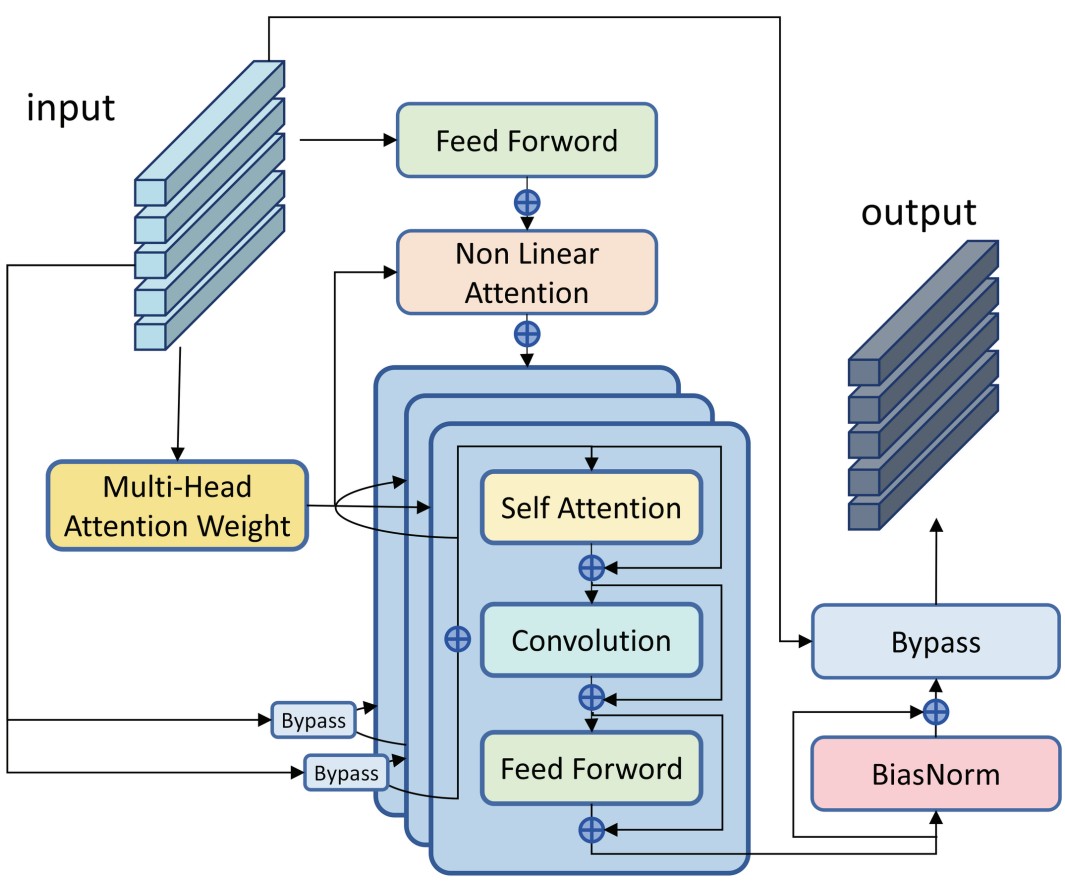

**Fig 2. Structure of the Improved Zipformer Block Module.**

Meanwhile, the speech features are also fed into the Feed Forward module, processed by the NLA module, and then passed through three consecutive SCF module groups. Finally, the output is normalized by the BiasNorm module.

The MHAW module maps the input embedding to Query, Key and relative position vectors through linear projection. It calculates the attention scores using matrix multiplication, models the sequence position relationships by incorporating relative position embeddings, and normalizes the scores into attention weights via softmax, which are then used for downstream sequence modeling tasks. The process can be described as follows:

$$Weight = Softmax(Q \cdot K^T + P \cdot Linear(pos\_emb) + Masking) \tag{1}$$

Where $Q$, $K$ are the query and key obtained by linear projection of the input tensor, respectively. $pos\_emb$ is the relative position encoding with shape (batch_size, 2*seq_len-1, embed_dim). $Linear(pos\_emb)$ is a linear transformation that maps the original relative position embedding to the required dimension of each attention head, providing independent position information for each head computation. $Masking$ is used to mask invalid positions, preventing the model from focusing on information that fills the region or future time steps.

The weights computed through the MHAW module are fed into the Non Linear Attrntion module and the SCF module groups, respectively. In this case, the Non Linear Attention

module aggregates the embedding vectors on the time axis using the attention weight values computed by MHAW. The module splits the input features into three parts by linear projection: $s$, $x$ and $y$. $s$ is input to the tanh activation function after Balancer processing, and $x$ is normalised and multiplied with the processed $s$ at the element level. Subsequently, the features are weighted by pre-computed attention weights to complete the aggregation of sequence information. The aggregated features are recovered to the input dimension by linear projection and normalised to generate the final output features. The specific formula is as follows:

$$X_{out} = Whiten(ScaledLinear(A * (Whiten(x) \odot tanh(Balancer(s))) \odot y)) \tag{2}$$

Where, *whiten* is a data normalisation technique used to reduce the correlation between the input features. *ScaledLinear* is a scaling based linear transformation layer. *tanh* is a nonlinear activation function. *Balancer* is used to limit the activation values to prevent training instability.

In this paper, groups of three serially connected SCF modules are cascaded, and a Bypass module is embedded between each group of SCF modules with additional inputs of MHAW-calculated Attention Weights and Non Linear Attention output values. The Bypass module is specifically described as:

$$Bypass(x, y) = (1 - c) \odot x + c \odot y \tag{3}$$

where $x$ is the input value of the previous module, $y$ is the output value of the previous module, and $c$ is the learning channel scalar weight.

The SCF module group includes Self Attention, Convolution and Feed Forword respectively. Fig 3 shows Self Attention. The module requires input features $X$ and attention weights $W$. Firstly, the features $X$ are mapped to $X_{linear}$ through a linear layer, after which the attention weights $W$ and $X_{linear}$ are computed by matrix multiplication. The resulting results are processed through *Whiten* in order to adjust the covariance of the features during backpropagation to be closer to the unit matrix for better feature decorrelation and improved generalisation of the model. Convolution uses a standard one-dimensional deep convolution module with symmetric padding at the end of the sequence in order to keep the output size constant. Feed Forword uses a feed-forward neural network layer to achieve deep processing and decorrelation of input features.

## 3.2 GELU-based pred network module

RNN-T models perform well in tasks such as speech recognition, but the complexity and number of parameters of their model structure for processing labels is large, leading to difficulties when applied on resource-constrained devices. To address the challenge of reducing the model size in RNN-T architectures without significantly degrading the recognition performance, Ghodsi et al. proposed the stateless Pred Network [23], which simplifies the model structure by removing the recurrent layer in the Pred Network while maintaining the performance of the model. However, the limitation of neuron deactivation occurs when the inputs to the ReLu activation function in this model are negative, i.e., the outputs are forced to zero, which restricts its ability to handle complex data patterns. In order to solve this problem, this paper improves the stateless Pred Network by introducing the GELU (Gaussian Error Linear Unit) activation function, which makes the activation function more sensitive to

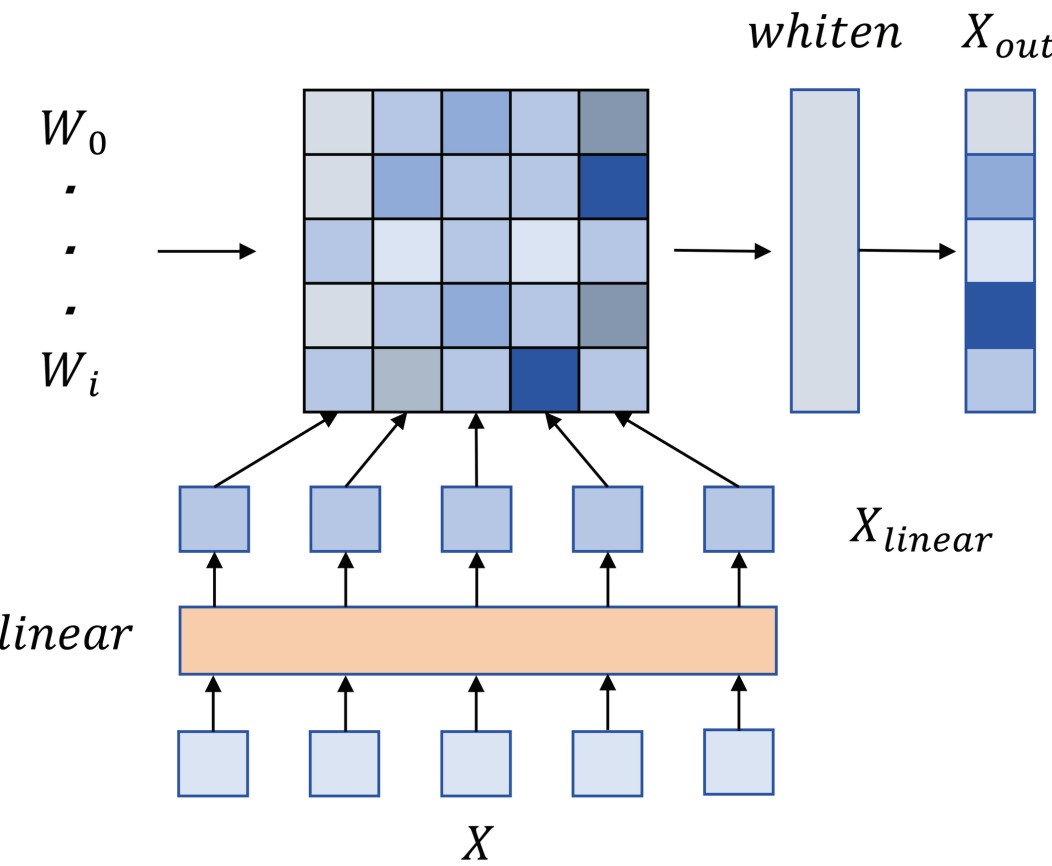

**Fig 3. Structure of Self Attention module using input weights in this paper.**

changes in the inputs, and exhibits better performance and generalisation ability when dealing with complex high-dimensional data.

Fig 4 shows the structure of the improved stateless Pred network, transform text label sequences into a continuous feature space through embedding layers, the Balancer is used to control the distribution of activation values to avoid the negative impact of extreme values on the model training, and then a one-dimensional convolutional layer is used to capture local information in the sequence, and the GELU activation function is used to introduce the non-linearities to help the model extract more complex features. The GELU activation function is used to introduce nonlinearities and help the model extract more complex features. The activated feature vector is passed through the Balancer layer again to further adjust the range of values in the output, and the final output is a vector with the shape of (N, U, decoder_dim). This structure ensures that the output vector captures the features of the input text while remaining numerically stable.

The GELU activation function provides a smoother activation curve and an adaptive gating mechanism, which makes the activation function more sensitive to changes in the input and reduces the instability of the gradient during the training process. The GELU activation function formula is as follows:

$$GELU(x) = xP(X \leq x) = x\phi(x) = x \cdot \frac{1}{2}\left[1 + erf\left(\frac{x}{\sqrt{2}}\right)\right] \qquad (4)$$

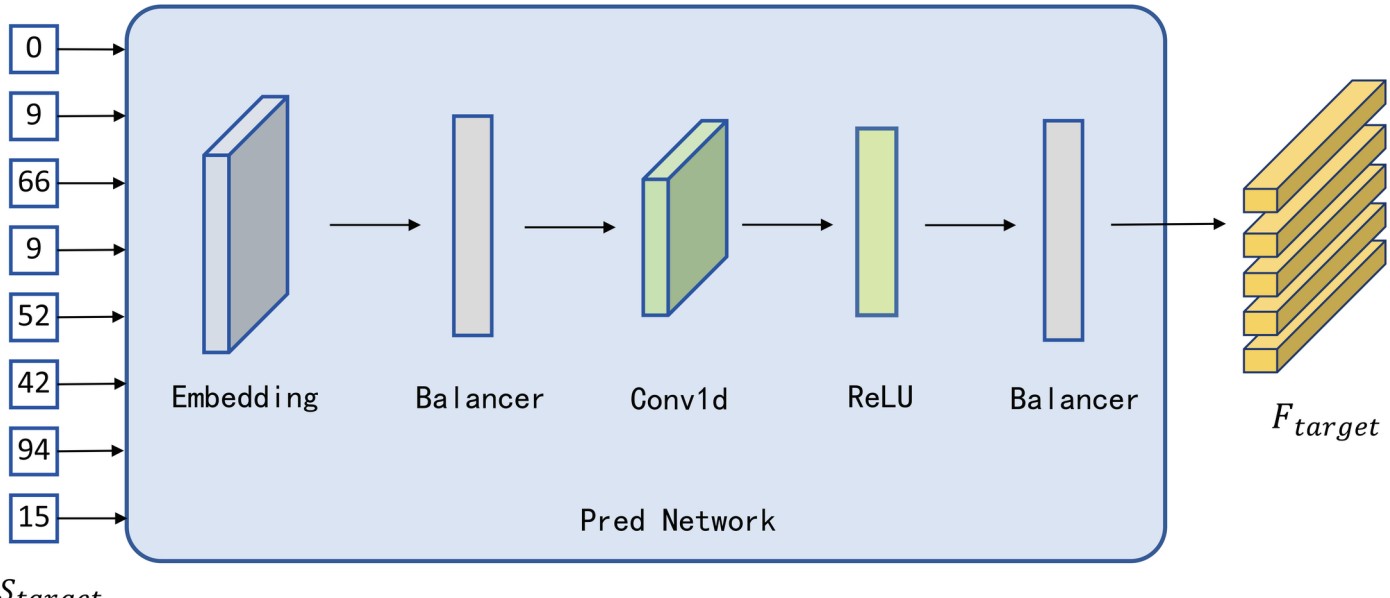

**Fig 4. Improved Pred Network module structure.**

Where $erf(\frac{x}{\sqrt{2}})$ is the error function.

The image of the GELU activation function is shown in Fig 5, which exhibits a smooth S-shaped curve that gradually tends to zero in the negative region and gradually increases linearly in the positive region, which is more natural and continuous compared to ReLU's segmented linearity and hard thresholding (where the negative values go directly to zero). This smoothness allows the model to be more delicate with inputs close to zero, avoids the complete neglect of negative values in ReLU, and is more stable during gradient propagation, thus aiding the optimisation process and the learning of complex features.

It has been proved by previous experiments that the GELU activation function can show better performance and generalisation ability when dealing with complex high-dimensional data, and it has been widely used in natural language processing models such as BERT (Bidirectional Encoder Representations from Transformers), GPT (Generative Pre-trained Transformer), GPT (Generative Pre-trained), XLNeT (Generalised Autoregressive Pretraining for Language Understanding), Transformer and other natural language processing models.

### 3.3 Mixed pruned RNN-T/CTC loss

The loss function is a key measure of the gap between the model's predicted results and the true results, and is a crucial part of optimising the model's performance.The Zipformer model introduces the Pruned RNN-T method proposed by Kuang et al. in the module of calculating the loss function [24], but the pruning strategy used in this method discards some information that is crucial in the alignment process, which reduces the model's accuracy and generalisation ability. To solve these problems, this paper introduces the CTC loss function on top of the Pruned RNN-T loss function, and proposes the hybrid Pruned RNN-T/CTC loss function method, which combines the Pruned RNN-T loss function and the CTC loss function in a weighted manner and balances the effects between them by adjusting the weights, aiming

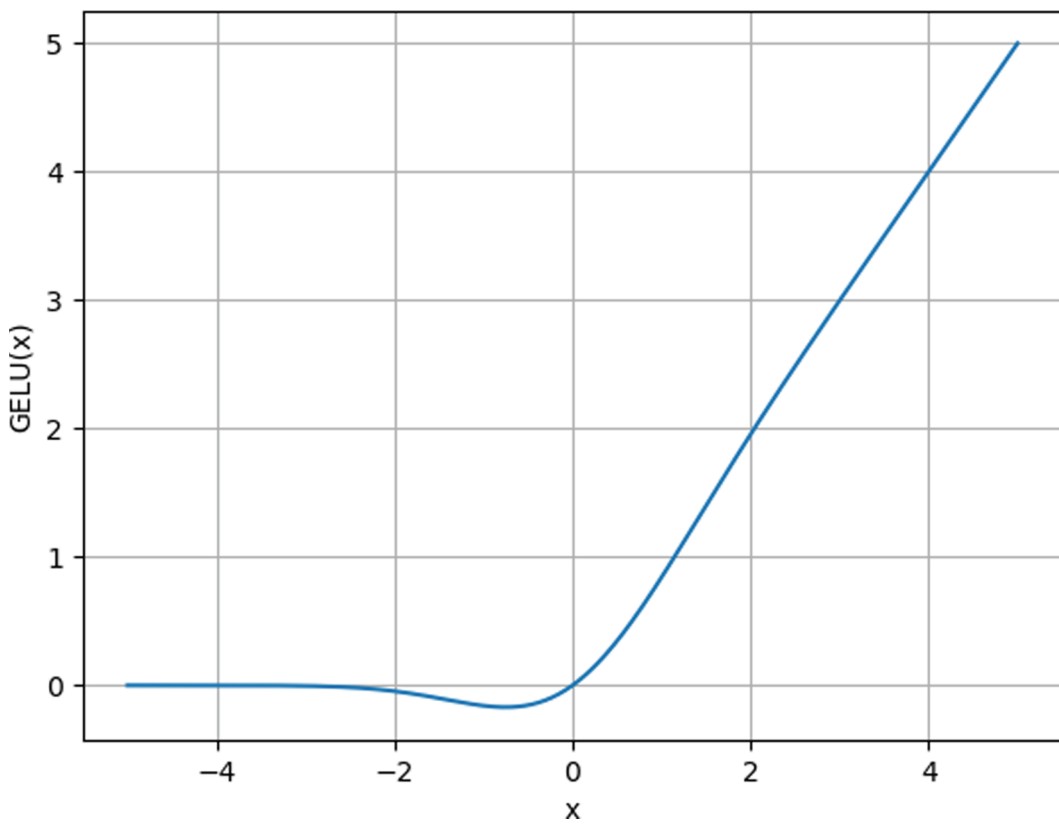

**Fig 5. Improved Pred Network module structure.**

to improve the performance of the model by combining multiple different loss functions through the combination to improve the performance of the model.

The overall structure of the hybrid Pruned RNN-T/CTC Loss is shown in Fig 6, and the overall structure is divided into the two loss functions of Pruned RNN-T and the CTC loss function. The Pruned RNN-T loss values include the Simple Loss and the Pruned Loss, and the Simple Loss refers to the loss value of the model before the pruning operation, based on its complete structure and all parameters, whereas Pruned Loss is the loss value recalculated after the model has been pruned, i.e., after some of the network parameters and/or nodes have been discarded to reduce model complexity and computation. In calculating the CTC loss, the encoder output is first mapped to a space more suitable for CTC through a linear layer, followed by a Dropout layer to reduce the overfitting and enhance the model generalisation by randomly dropping some of the outputs. Afterwards, the LogSoftmax layer converts these vectors into log-probability form, ensuring that each dimension represents the category log-probability and sums to 0. Finally, these log-probabilities and labels are subjected to CTC loss calculation.

The Hybrid Pruned RNN-T/CTC Loss is derived by weighted summation of the above three loss values and is given by the following formula:

$$L_{total} = W_{pruned} \cdot L_{pruned} + W_{simple} \cdot L_{simple} + 0.085 \cdot L_{ctc} \tag{5}$$

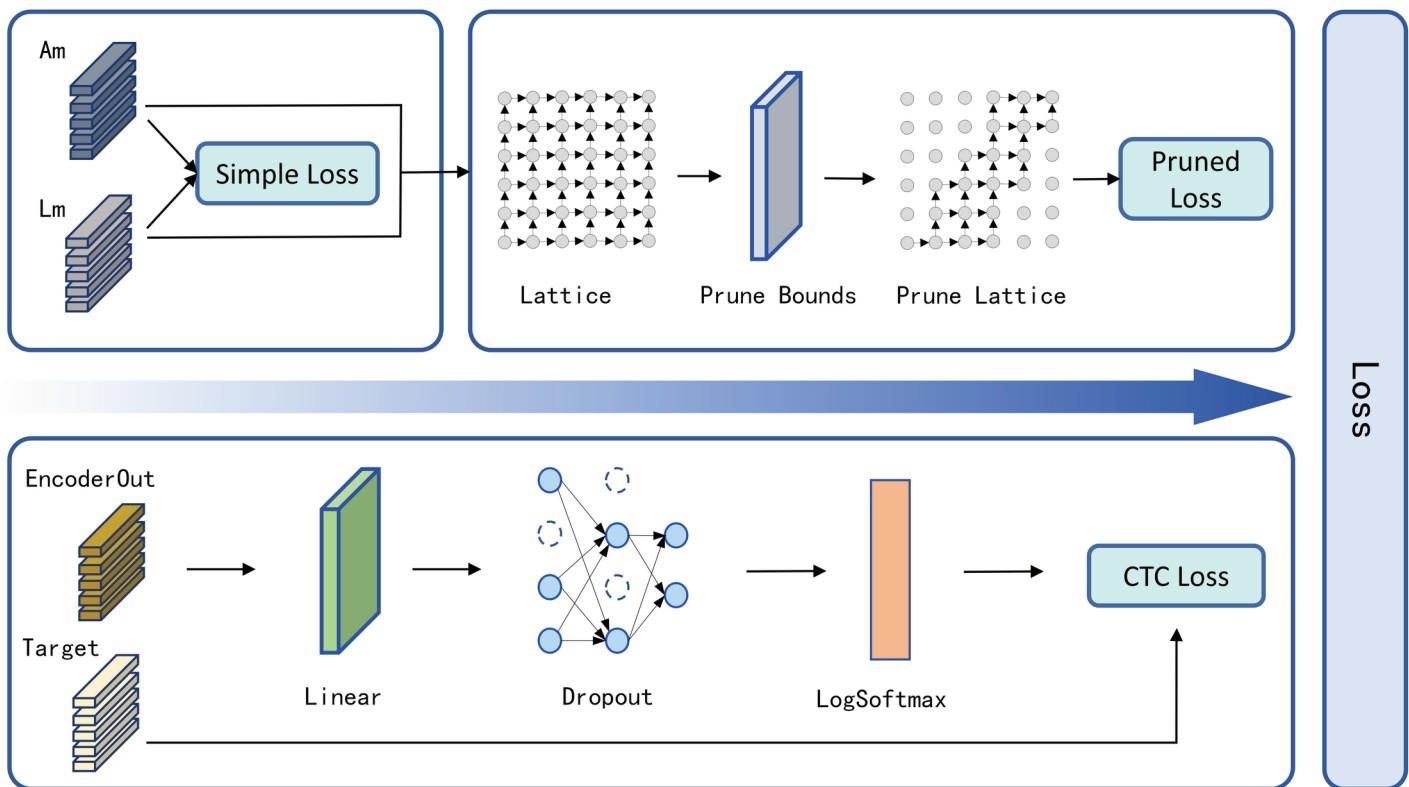

**Fig 6. Improved Pred Network module structure.**

Where $L_{total}$ is the total loss value, $L_{pruned}$, $L_{simple}$ and $L_{ctc}$ are the Pruned loss value, the Simple loss value and the CTC loss value, respectively, $W_{pruned}$ and $W_{simple}$ are the scaling factors for the Pruned loss value and the Simple loss value, respectively.

The calculation formulas for the two scaling factors are as follows:

$$W_{pruned} = \begin{cases} 1.0, & if\ batch\_idx \geq warm\_step \\ 0.1 + 0.9 \cdot \frac{batch\_idx}{warm\_step}, & otherwise \end{cases} \quad (6)$$

$$W_{simple} = \begin{cases} 0.65, & if\ batch\_idx \geq warm\_step \\ 1.0 - \frac{batch\_idx}{warm\_step}, & otherwise \end{cases} \quad (7)$$

Where $batch\_idx$ is the current training batch index, and $warm\_step$ is the step threshold.

As training progresses, the scaling factors of the loss function gradually transition from their initial values to the predetermined target values, enabling dynamic adjustment of the loss. Specifically, and are adjusted according to a linear progression during the training process, thus assigning appropriate weights to the smooth loss and pruning loss at different stages of training. Experimental results show that when the weight coefficient of the CTC loss is set to 0.085, the model achieves optimal overall performance. This approach effectively balances the influence of different loss components, significantly improving the model's recognition accuracy and stability.

## 4 Experimental results and analysis

### 4.1 Experimental data

The experiments in this study primarily use the Mandarin speech datasets AISHELL1 and ST-CMDS. The AISHELL1 dataset was originally recorded by 400 speakers from different regions of China using a variety of devices, including high-fidelity microphones, Android phones, and iOS phones, with an audio sampling rate of 16 kHz. The ST-CMDS dataset is an open-source Mandarin Chinese speech dataset primarily used for tasks such as speech recognition, speech synthesis, and speaker recognition. The dataset contains a large number of mono-phonic audio files with a 16 kHz sampling rate, covering various daily conversation scenarios, such as online chatting and voice command for smart devices.

To adapt the dataset for phoneme recognition tasks, this study performed a phoneme label conversion on the original Chinese character text labels, aiming to construct the AISHELL1-PHONEME and ST-CMDS-PHONEME Chinese phoneme datasets. The core step in this process was to replace the original Chinese character labels with phoneme labels, utilizing the mapping relationship between Chinese characters and phonemes. Due to the phenomenon of polyphonic characters in Chinese, where the pronunciation of a character may differ in different word groups or contexts, a simple character-by-character replacement approach cannot meet the need for accurate labeling. Therefore, this study employed a phrase-level phoneme mapping table to replace the labels, thereby avoiding the phonetic ambiguity that arises from character-by-character substitution of polyphonic characters. This method allows the dataset to more accurately reflect the phonetic characteristics of phonemes in different contexts, adapting to more fine-grained, phoneme-level speech recognition tasks. The dataset divisions for AISHELL1-PHONEME and ST-CMDS-PHONEME are shown in Tables 1 and 2.

To enhance the robustness of the model, this paper mixes the MUSAN dataset into the AISHELL1-PHONEME dataset to simulate noise interference in real-world environments, thereby improving the model's adaptability under various noise conditions. The MUSAN dataset is a comprehensive corpus that includes various types of audio data, such as music, speech, and noise. These audio data come from multiple sources and cover a wide range of noise scenarios, including background noise, traffic noise, wind noise, animal sounds, as well as various types of music and speech materials, greatly enriching the dataset's diversity.

By combining the MUSAN dataset with the AISHELL1-PHONEME dataset, various noise interference situations closer to real-world environments can be simulated, including noise from different frequency bands, dynamic noise, and sudden bursts of noise. Introducing this

**Table 1. The division of the AISHELL1-PHONEME dataset.**

| Data | Time(h) | Phoneme Quantity |
|------|---------|------------------|
| Train | 150 | 10365415 |
| Dev | 18 | 410674 |
| Test | 10 | 209524 |

**Table 2. The division of the ST-CMDS-PHONEME dataset.**

| Data | Time(h) | Phoneme Quantity |
|------|---------|------------------|
| Train | 85 | 1900432 |
| Dev | 10 | 223750 |
| Test | 5 | 111618 |

noise during the training process can effectively improve the model's robustness, enhancing its ability to adapt to different environmental noises, allowing it to better handle variable and complex speech inputs in practical applications. Table 3 shows the classification of the MUSAN dataset.

## 4.2 Evaluation metrics

The experiments in this paper use three metrics to evaluate the model's performance and efficiency: Word Error Rate (WER), Real Time Factor (RTF), and the number of parameters (Params).

**4.2.1 WER.** Word Error Rate (WER) is a key metric for measuring the degree of difference between the output phonemes of a speech recognition model and the reference phonemes. It is used to evaluate the accuracy of a speech recognition model in transcribing speech. Specifically, WER quantifies the accuracy of the model under various conditions by calculating the difference between the model's output and the true labels (reference phonemes). It considers three basic types of errors: substitution (Sub), insertion (Ins), and deletion (Del), providing a comprehensive reflection of the potential biases that may occur during the recognition process.

The calculation method of WER normalizes the total occurrences of substitution, insertion, and deletion errors by the total number of reference phonemes, yielding a ratio. This ratio indicates the extent of errors made by the model when processing a segment of speech. The lower the WER, the higher the model's recognition accuracy, as it better matches the true phoneme labels. The calculation formula for WER is typically as follows:

$$WER = \frac{Sub + Ins + Del}{Num} * 100 \tag{8}$$

Where *Sub* represents the number of substitution errors, which refers to the times the model's output phonemes do not match the reference phonemes; *Ins* represents the number of insertion errors, which refers to the number of extra or inserted phonemes in the model's output; *Del* represents the number of deletion errors, which refers to the number of phonemes missing or deleted in the model's output. *Num* represents the total number of reference phonemes, which is the number of phonemes in the true labels.

By calculating the total of these error types and normalizing it with the total number of reference phonemes, WER provides a quantitative assessment of the accuracy of the speech recognition model. It is important to note that WER not only considers the model's phoneme recognition accuracy but also reflects its performance in handling complex cases such as homophones, polyphonic characters, and noise interference. Therefore, WER is widely used as a standard to evaluate the performance of speech recognition systems.

In practical applications, WER also has its limitations, especially in cases with long texts or complex contexts, where a single error type may not fully capture the model's performance.

**Table 3. The categorization of the MUSAN dataset.**

| Categories | Time(h) | Source |
|---|---|---|
| music | 42 | jazz music, classical music, pop music, etc. |
| speech | 60 | English, French, German, Spanish, etc. |
| noise | 6 | crowd noise, traffic noise, white noise, etc. |

As a result, in addition to WER, other metrics such as Real-Time Factor (RTF) and Parameters (Params) are often used in combination to comprehensively evaluate the model's recognition performance and operational efficiency.

**4.2.2 RTF.** Real-Time Factor (RTF) is one of the key metrics used to evaluate the real-time performance and processing speed of speech recognition models, particularly in applications that require fast response times. The real-time factor represents the ratio between the time taken by the model to process a segment of audio and the actual duration of that audio. In the domain of phoneme recognition, RTF is used to measure how much audio the model can process within a given unit of time. It is typically employed to assess the response speed and processing capability of a speech recognition model when handling real-time audio data.

Specifically, the real-time factor quantifies the processing efficiency of a model by calculating the ratio between the time taken by the model to process audio data and the actual duration of the input audio. When the real-time factor is less than 1, it indicates that the model can process audio in real-time (i.e., within the same duration as the input), meaning the model processes more than 1 second of audio per second, which typically implies high processing speed and low latency. Conversely, when the real-time factor exceeds 1, it suggests that the model's processing speed cannot meet real-time requirements, as the time taken exceeds the duration of the audio, resulting in increased latency and making the model unsuitable for real-time applications. Therefore, the smaller the real-time factor, the faster the processing speed and the shorter the response time, making it more suitable for practical applications that require rapid responses, such as voice assistants, telephone speech recognition, and intelligent customer service. The formula for calculating the real-time factor is as follows:

$$RTF = \frac{InfertenceTime}{AudioDuration} \tag{9}$$

Where *Inference Time* refers to the time taken by the model to process the input audio, typically measured in seconds; and *Audio Duration* refers to the duration of the input audio data, also typically measured in seconds.

A low Real-Time Factor (RTF) is crucial for the practical application of speech recognition systems, especially in edge devices or resource-constrained environments. To optimize the real-time factor, it is typically necessary to improve the computational efficiency of algorithms, leverage hardware acceleration, and reduce the complexity of the model to enhance the system's response speed. In the design of speech recognition systems, the real-time factor needs to be balanced with other metrics (such as accuracy, model size, etc.) to ensure that the system can process data efficiently while maintaining good accuracy and reliability.

**4.2.3 Params.** The number of parameters(Params) is one of the key indicators of model complexity and expressive capability. In speech recognition tasks, particularly phoneme-level recognition, the number of parameters plays a crucial role in the model's performance. A higher number of parameters enables the model to have a stronger fitting ability, allowing it to capture more complex speech features and thus improve recognition accuracy.

However, more parameters do not mean better model performance. Although increasing the number of parameters can enhance the model's performance, an excessively high number of parameters also brings a range of negative impacts. First, an overly large number of parameters can lead to overfitting, where the model performs well on training data but poorly on new data (especially test data or real-world data). Overfitting occurs when the model learns too much noise and unnecessary details during training, losing its ability to generalize and thus resulting in decreased performance. Second, an excessive number of parameters significantly increases the computational resources and memory consumption of the model. Each

learnable parameter must be stored and computed during both training and inference, which directly leads to increased computation time. This is particularly problematic when processing large-scale speech data, where the computational and storage costs can be very high. Additionally, as the number of parameters increases, the training time also grows significantly, prolonging the development cycle. More importantly, during real-world deployment, especially in resource-constrained scenarios such as edge devices or embedded systems, models with an excessively large number of parameters increase deployment costs and complexity. This may require stronger hardware support, larger storage space, and higher computational power, which is unacceptable in many real-time application scenarios.

Therefore, when designing phoneme recognition models, it is essential to control the number of parameters appropriately. Ensuring high recognition accuracy while maintaining a low model complexity is one of the key strategies for optimizing model performance and resource utilization. Moreover, choosing an appropriate model size for different application scenarios is also an important aspect of model optimization. For instance, in embedded systems or mobile devices, smaller models may be required to balance performance, accuracy, and resource consumption.

## 4.3 Model validation

**4.3.1 Comparison of mainstream models.** This set of experiments adopts the state-of-the-art encoder-decoder architecture to thoroughly evaluate the performance of three mainstream acoustic models—Zipformer, Conformer, and Transformer—on the Mandarin phoneme recognition task. The experiments are based on the AISHELL1-PHONEME and ST-CMDS-PHONEME datasets, using the Fbank technique to extract speech features. Eight different phoneme recognition models are trained, and their performance is evaluated using word error rate (WER), real-time factor (RTF), and model parameter count from both the development and test sets. In the experiments, Zipformer-RNN-T(Pruned) demonstrates excellent performance on both the AISHELL1-PHONEME and ST-CMDS-PHONEME datasets, exhibiting strong robustness and high generalization ability.

As shown in Table 4, on the AISHELL1-PHONEME dataset, Zipformer-RNN-T(Pruned) achieves a Dev WER of 2.28% and a Test WER of 2.12%, significantly outperforming other models. In particular, compared to Conformer-CTC (Test WER 2.88%), Transformer-CTC (Test WER 5.78%), and Zipformer-CTC (Test WER 2.80%), Zipformer-RNN-T(Pruned) significantly reduces the error rate on the test set. Additionally, Zipformer-RNN-T(Pruned) has an inference speed (RTF) of 0.002, which is significantly faster than Conformer-RNN-T (RTF of 0.012) and Transformer-RNN-T(Pruned) (RTF of 0.003), demonstrating a clear advantage in inference efficiency.

As shown in Table 5, on the ST-CMDS-PHONEME dataset, Zipformer-RNN-T(Pruned) achieves a Dev WER of 4.28% and a Test WER of 4.51%, outperforming all other comparison models. Especially when compared to Transformer-CTC (Test WER 15.77%), Conformer-CTC (Test WER 6.47%), and Zipformer-CTC (Test WER 6.43%), Zipformer-RNN-T(Pruned) performs excellently in terms of accuracy. Although its Dev WER (4.81%) and Test WER (4.97%) are not significantly different from those of Conformer-RNN-T(Pruned), Zipformer-RNN-T(Pruned) still holds an advantage in inference speed (RTF of 0.002) and model size (61.1M parameters), demonstrating a balanced trade-off between efficiency and performance.

These results indicate that Zipformer-RNN-T(Pruned) achieves optimal performance on both the AISHELL1-PHONEME and ST-CMDS-PHONEME datasets, with the fewest parameters, lowest error rate, and fastest inference speed, showcasing its superior robustness and generalization ability in phoneme recognition tasks. Compared to all other models,

**Table 4. Comparative experimental results of mainstream models on the AISHELL1-PHONEME dataset.**

| Model | Dev | Test | RTF | Params |
|---|---|---|---|---|
| Conformer-CTC | 2.53% | 2.88% | 0.029 | 108.7 |
| Transformer-CTC | 5.03% | 5.78% | 0.020 | 70.7 |
| Zipformer-CTC | 2.46% | 2.80% | 0.018 | 90.4 |
| Zipformer-RNN-T | 2.28% | 2.45% | 0.010 | 67.4 |
| Conformer-RNN-T | 2.44% | 2.71% | 0.012 | 83.7 |
| Transformer-RNN-T(Pruned) | 3.97% | 4.53% | 0.003 | 66.4 |
| Conformer-RNN-T(Pruned) | 2.21% | 2.51% | 0.003 | 78.1 |
| Zipformer-RNN-T(Pruned) | **1.92%** | **2.12%** | **0.002** | **61.1** |

**Table 5. Comparative Experimental Results of Mainstream Models on ST-CMDS-PHONEME Dataset.**

| Model | Dev | Test | RTF | Params |
|---|---|---|---|---|
| Conformer-CTC | 6.22% | 6.47% | 0.029 | 108.7 |
| Transformer-CTC | 15.02% | 15.77% | 0.020 | 70.7 |
| Zipformer-CTC | 6.10% | 6.43% | 0.018 | 90.4 |
| Zipformer-RNN-T | 5.74% | 5.94% | 0.010 | 67.4 |
| Conformer-RNN-T | 4.98% | 5.11% | 0.011 | 83.7 |
| Transformer-RNN-T(Pruned) | 11.73% | 12.38% | 0.003 | 66.4 |
| Conformer-RNN-T(Pruned) | 4.81% | 4.97% | 0.003 | 78.1 |
| Zipformer-RNN-T(Pruned) | **4.28%** | **4.51%** | **0.002** | **61.1** |

Zipformer-RNN-T(Pruned) leads across all evaluation metrics, proving its advantage as the optimal model.

In the phoneme recognition task, misclassifications are typically categorized into three types: substitution, insertion, and deletion. As shown in Figs 7 and 8, on both the AISHELL1-PHONEME and ST-CMDS-PHONEME datasets, the number of substitution errors is significantly higher than that of insertion and deletion errors. In the specific misclassification statistics, Zipformer-RNN-T(Pruned) achieves the lowest error count across all error types. On the AISHELL1-PHONEME dataset, the numbers of substitution, insertion, and deletion errors are 7165, 380, and 356, respectively; on the ST-CMDS-PHONEME dataset, they are 7890, 741, and 820. Compared to other models, Zipformer-RNN-T(Pruned) performs exceptionally well in reducing misclassification counts, demonstrating strong modeling capability and stability both in handling the dominant substitution errors and the relatively less frequent insertion and deletion errors.

According to the phoneme dictionary provided by the AISHELL1 dataset, Mandarin pronunciation is subdivided into 66 distinct phonemes. As shown in Figs 9 and 10, the proposed model Zipformer-RNN-T(Pruned) demonstrates good phoneme recognition accuracy on both the AISHELL1-PHONEME and ST-CMDS-PHONEME datasets, covering all phoneme categories. The accuracy for the majority of phonemes is higher than that of other comparative models.

**4.3.2 Noise simulation experiment.** To enhance the model's generalization ability and robustness, this study conducts a data augmentation experiment. The MUSAN dataset is used as the background noise source and mixed with the original Train, Dev, and Test sets of the AISHELL1-PHONEME and ST-CMDS-PHONEME datasets, generating augmented datasets: Train+, Dev+, and Test+. The experiment adopts a dual-track parallel training strategy, where phoneme recognition models are trained based on both the Train+ and Train datasets, and

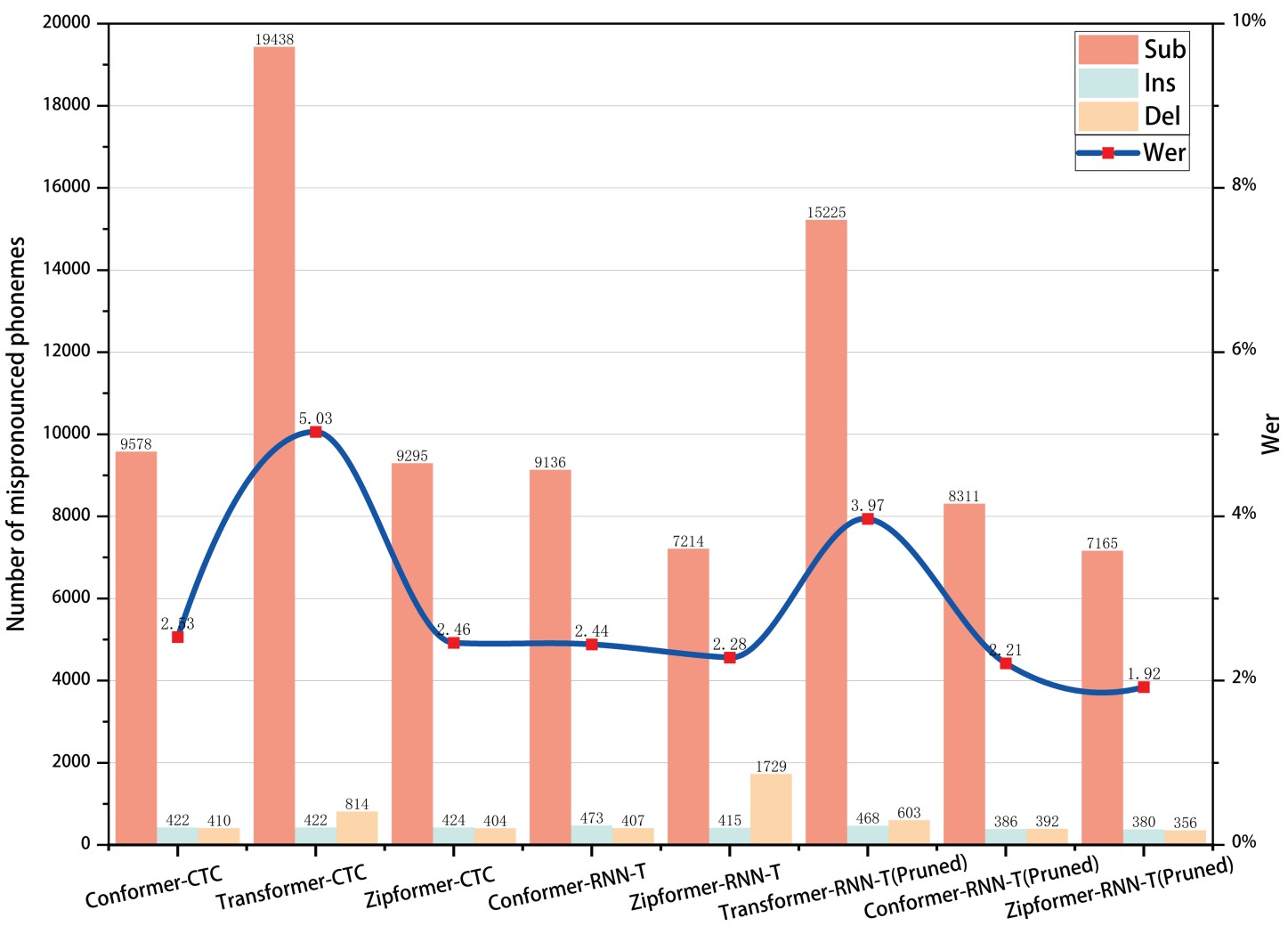

**Fig 7. Word error rates and the number of three types of misrecognized phonemes on the development set for different models(AISHELL1-PHONEME).**

validated and tested on the Dev+ and Test+ datasets. As shown in Tables 6 and 7, the models trained on Train+ outperform those trained on Train in both Word Error Rate (WER) and Real-Time Factor (RTF), indicating the effectiveness of the data augmentation strategy in improving model performance. This strategy not only strengthens the model's ability to accurately recognize speech content, but also significantly enhances its robustness when faced with noise interference. Additionally, the proposed Zipformer-RNN-T(Pruned) model demonstrates exceptional performance on both training sets in the AISHELL1-PHONEME and ST-CMDS-PHONEME datasets, with significantly lower WER and RTF compared to other comparative models. This highlights the model's strong robustness and ability to maintain stable performance in complex and dynamic acoustic environments.

## 4.4 Ablation experiment

In phoneme recognition tasks, the misalignment between the input speech features and the output phoneme sequence is a common issue. The CTC (Connectionist Temporal

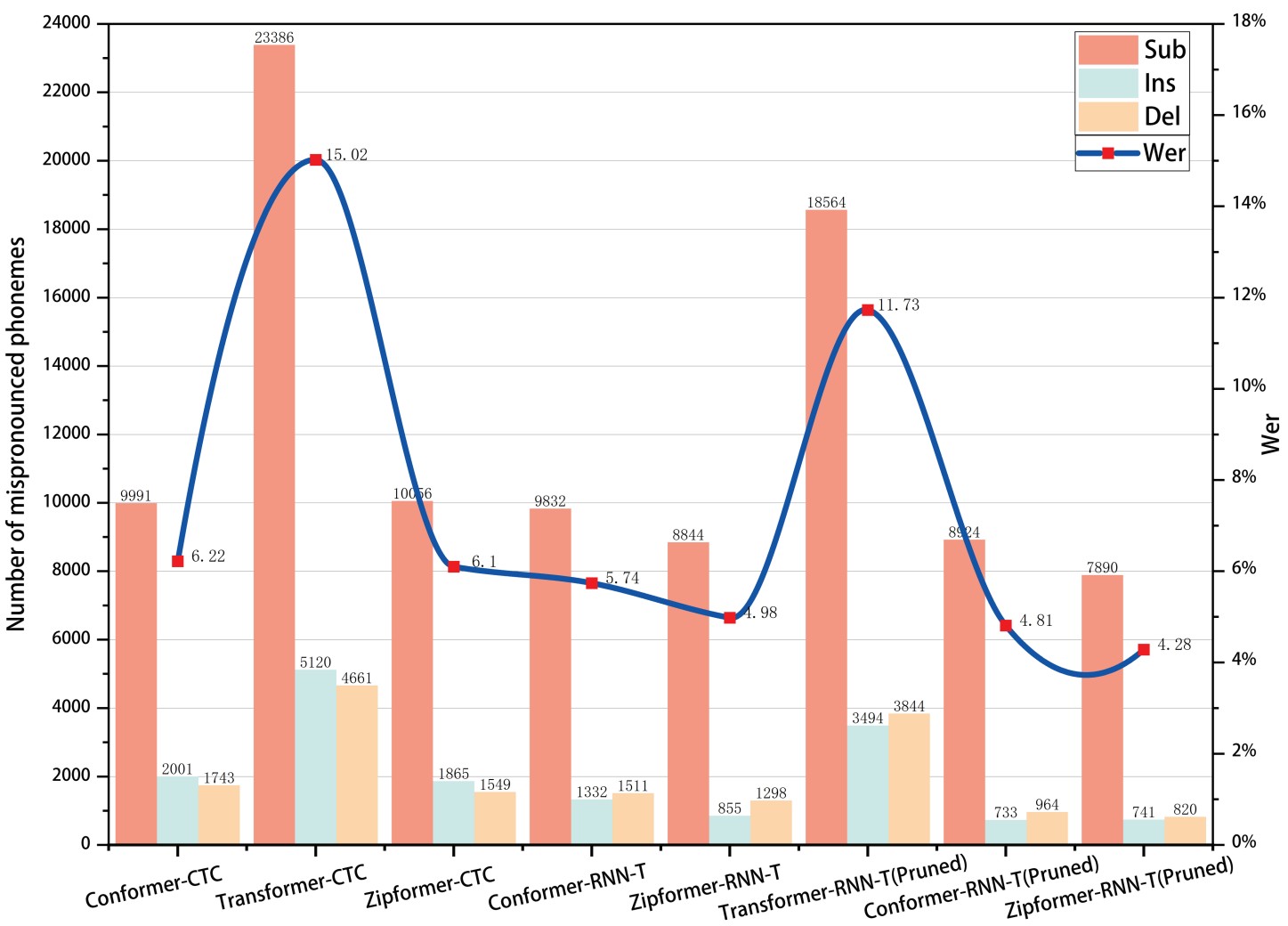

**Fig 8. Word error rates and the number of three types of misrecognized phonemes on the development set for different models(ST-CMDS-PHONEME).**

Classification) loss function addresses this problem by introducing a blank symbol and allowing multiple possible alignment paths, which enables automatic alignment between the input and output sequences. This improves the model's robustness and generalization ability. The weight configuration of the CTC loss function is an essential part of model performance tuning, and setting the appropriate weight can further enhance the accuracy of phoneme recognition.

In the experiment of adjusting the CTC weight coefficient, the model was first trained on the Train+ dataset to ensure that it could sufficiently learn the mapping between speech features and phoneme labels in the standard training set. Then, the mixed MUSAN dataset was applied to the Dev+ and Test+ datasets for testing, to evaluate the model's performance when noise interference was introduced. Fig 11 presents the experimental results showing the difference in model performance on the Dev+ and Test+ datasets with different CTC weighting factor settings.

When the CTC weight coefficient was set to 0.085, the model achieved the lowest Word Error Rate (WER) on both the Dev+ and Test+ datasets, with rates of 1.95% and 2.11%,

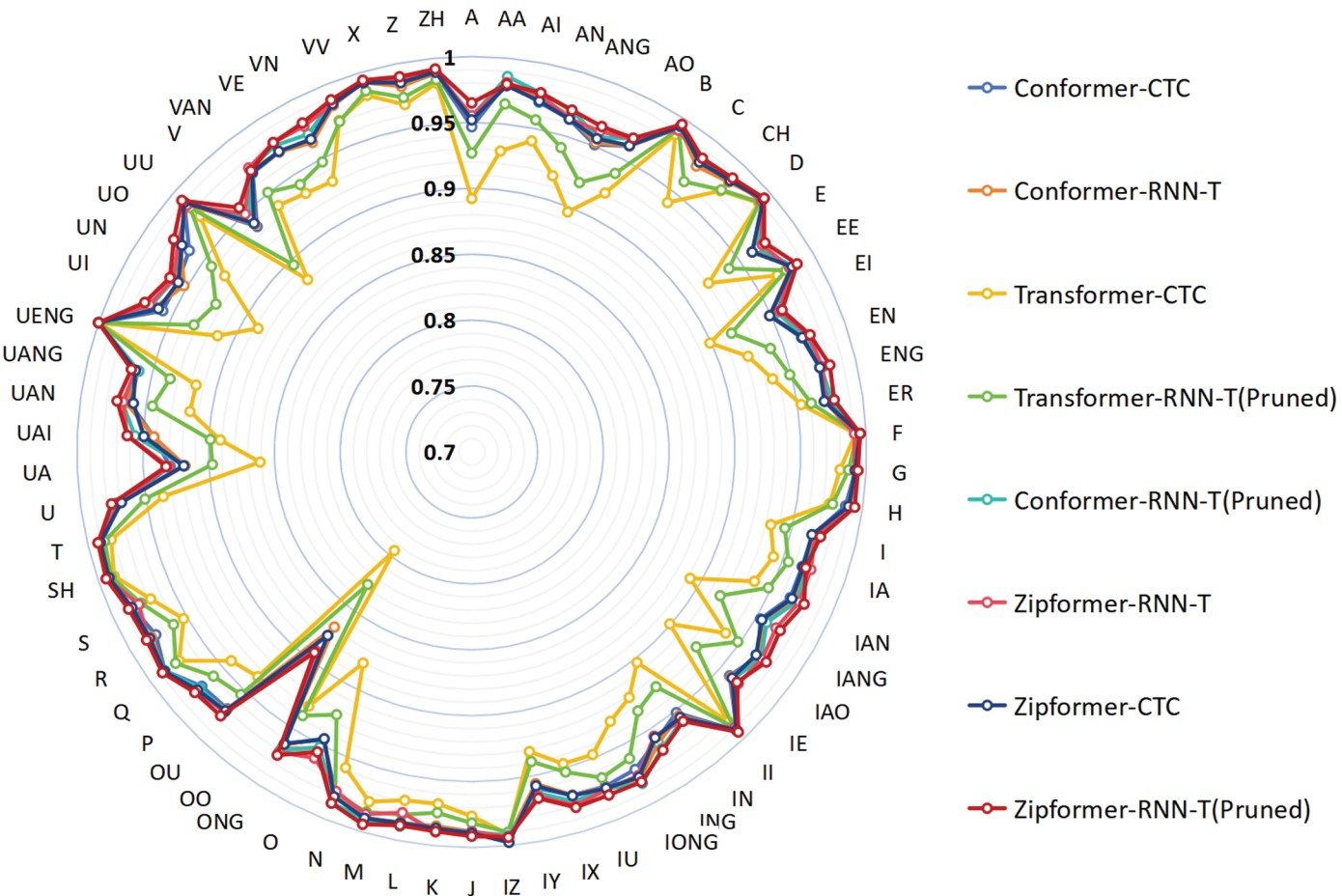

**Fig 9. The phoneme accuracy percentage on the development set for different models (AISHELL1-PHONEME).**

respectively. This result indicates that the setting of the CTC weight coefficient has a significant impact on the performance of the phoneme recognition task. In particular, in complex noisy environments, an appropriate weight coefficient can significantly improve the model's recognition accuracy and stability. Under this weight coefficient, the model was able to effectively capture phoneme information in noisy and misaligned speech data, thereby optimizing the accuracy of speech recognition.

When the CTC weight coefficient deviated from 0.085, the experimental results showed a gradual increase in WER, demonstrating a certain degree of performance degradation. This trend suggests that both excessively high and low CTC weight coefficients can negatively impact model training. For example, if the CTC weight coefficient is too high, the model may overly rely on the blank symbol, thus ignoring phoneme label information and leading to a decline in recognition accuracy. Conversely, if the coefficient is too low, the model may not be flexible enough in alignment, which negatively affects its ability to capture phonemes and leads to more errors. This gradually increasing WER trend further validates the role of the CTC weight coefficient in regulating the model's training and inference process. An appropriate weight coefficient helps the model better balance the selection of alignment paths,

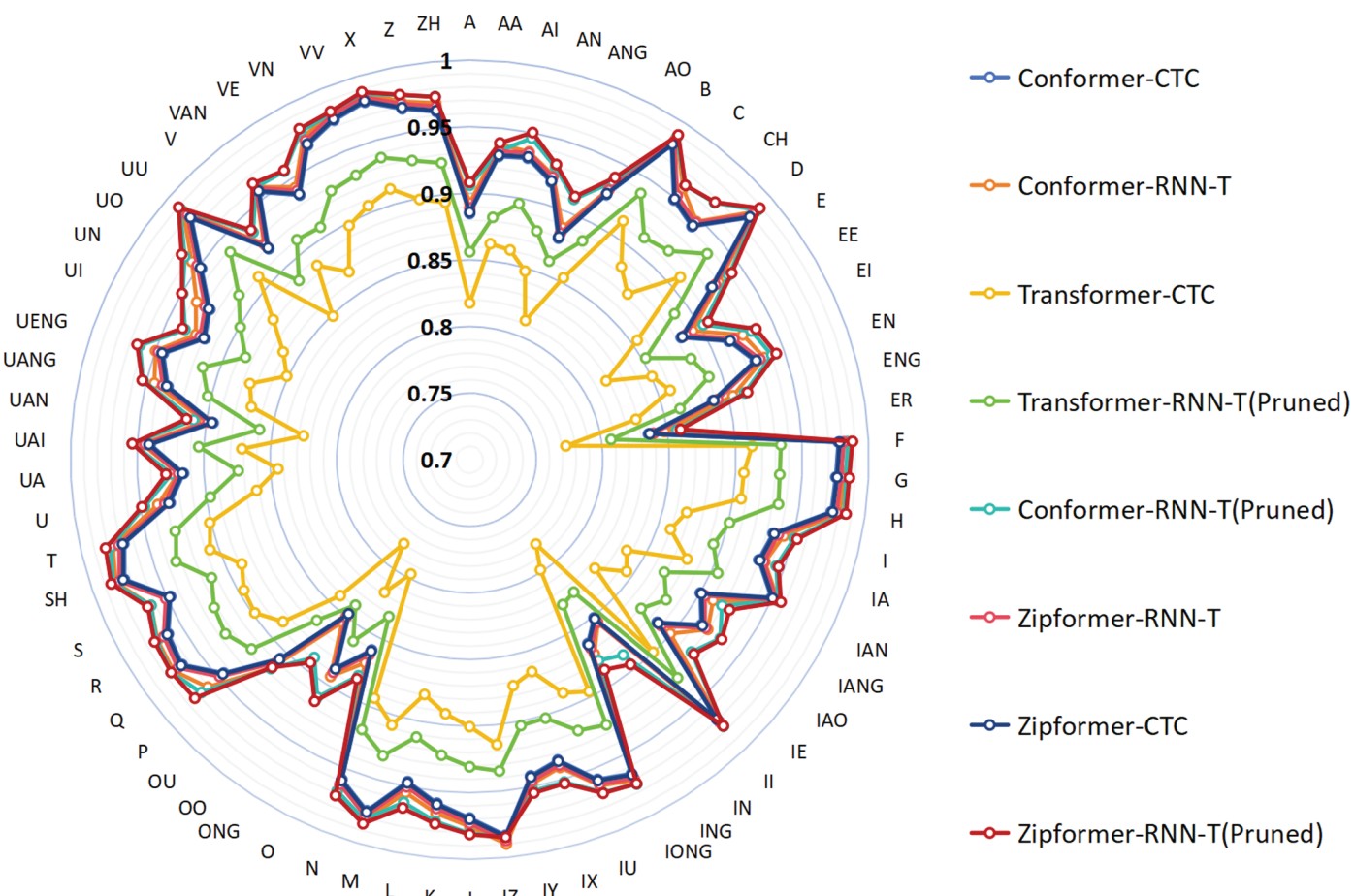

**Fig 10. The phoneme accuracy percentage on the development set for different models (ST-CMDS-PHONEME).**

Table 6. Comparison of Model Performance Under Different Data Augmentation Strategies on the AISHELL1-PHONEME Dataset.

| Model | Train+ | | | Train | | | Para |
|---|---|---|---|---|---|---|---|
| | Dev+ | Test+ | RTF | Dev+ | Test+ | RTF | |
| Conformer-CTC | 2.63% | 3.00% | 0.015 | 3.05% | 3.39% | 0.029 | 108.7 |
| Transformer-CTC | 5.33% | 6.04% | 0.020 | 6.14% | 6.87% | 0.020 | 70.7 |
| Zipformer-CTC | 2.65% | 2.97% | 0.018 | 2.87% | 3.17% | 0.018 | 90.4 |
| Zipformer-RNN-T | 2.47% | 2.79% | 0.005 | 2.92% | 3.15% | 0.005 | 67.4 |
| Conformer-RNN-T | 2.27% | 2.51% | 0.004 | 2.51% | 2.71% | 0.010 | 83.7 |
| Transformer-RNN-T(Pruned) | 4.38% | 5.03% | 0.003 | 4.88% | 5.44% | 0.003 | 66.4 |
| Conformer-RNN-T(Pruned) | 2.19% | 2.44% | 0.003 | 2.57% | 2.83% | 0.003 | 78.1 |
| Zipformer-RNN-T(Pruned) | **1.95%** | **2.11%** | **0.002** | **2.16%** | **2.34%** | **0.002** | **61.1** |

avoiding over-reliance on blank symbols or other unreasonable alignment methods, thus improving the accuracy of phoneme recognition.

To systematically assess the impact of each component on the overall performance of the model, this paper designs a series of ablation experiments based on the Zipformer-RNN-T(Pruned)

**Table 7. Comparison of Model Performance Under Different Data Augmentation Strategies on the ST-CMDS-PHONEME Dataset.**

| Model | Train+ | | | Train | | | Para |
|---|---|---|---|---|---|---|---|
| | Dev+ | Test+ | RTF | Dev+ | Test+ | RTF | |
| Conformer-CTC | 6.38% | 6.75% | 0.025 | 6.84% | 7.16% | 0.028 | 108.7 |
| Transformer-CTC | 15.37% | 15.99% | 0.022 | 16.21% | 16.93% | 0.021 | 70.7 |
| Zipformer-CTC | 6.44% | 6.78% | 0.016 | 6.72% | 6.97% | 0.016 | 90.4 |
| Zipformer-RNN-T | 5.88% | 6.15% | 0.005 | 6.25% | 6.48% | 0.005 | 67.4 |
| Conformer-RNN-T | 5.37% | 5.42% | 0.004 | 5.55% | 5.74% | 0.008 | 83.7 |
| Transformer-RNN-T(Pruned) | 12.55% | 13.18% | 0.005 | 12.97% | 13.49% | 0.005 | 66.4 |
| Conformer-RNN-T(Pruned) | 4.98% | 5.09% | 0.004 | 5.34% | 5.53% | 0.004 | 78.1 |
| **Zipformer-RNN-T(Pruned)** | **4.39%** | **4.54%** | **0.003** | **4.66%** | **4.90%** | **0.003** | **61.1** |

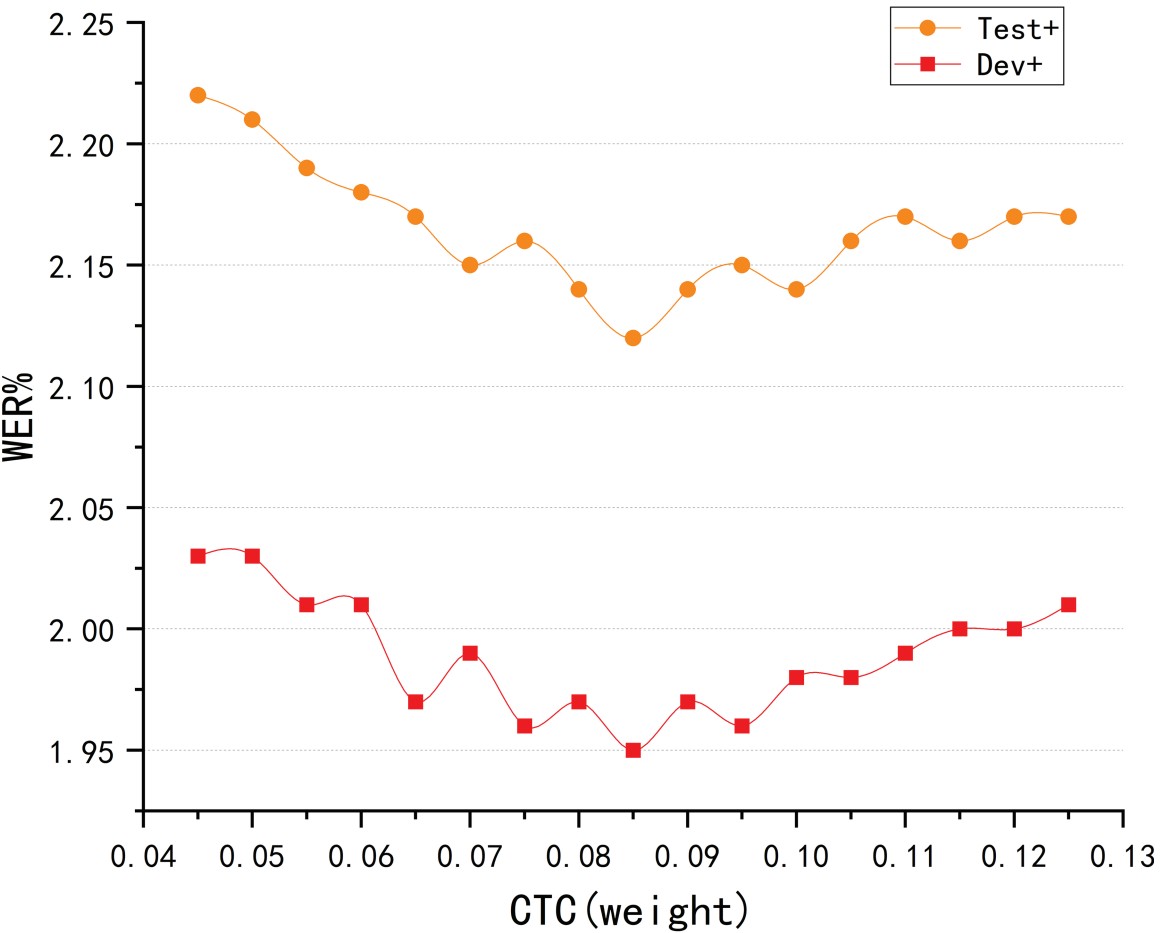

**Fig 11. The influence trend of different CTC weight coefficients on Word Error Rate (WER).**

architecture. The aim is to quantify the influence of different modules (such as the Block3 module, GELU activation function, and CTC loss function) on recognition accuracy (Word Error Rate, WER), real-time factor (RTF), and model complexity (parameter count, Params) by comparing the model's performance under different configurations.

**Table 8. Ablation experiment.**

| Model | Block3 | GELU | CTC | Dev+ | | Test+ | | Para |
|---|---|---|---|---|---|---|---|---|
| | | | | Wer | RTF | Wer | RTF | |
| Zipformer-RNN-T(Pruned)-L | | | | 2.04% | 0.002 | 2.23% | 0.002 | 65.1 |
| Zipformer-RNN-T(Pruned) | √ | √ | √ | **1.95%** | 0.002 | **2.11%** | 0.002 | 61.1 |
| Zipformer-RNN-T(Pruned) | | √ | √ | 1.99% | 0.002 | 2.12% | 0.002 | 61.1 |
| Zipformer-RNN-T(Pruned) | √ | | √ | 1.97% | 0.002 | 2.12% | 0.002 | 61.1 |
| Zipformer-RNN-T(Pruned) | √ | √ | | 1.98% | 0.002 | 2.16% | 0.002 | 61.1 |

First, the Zipformer-RNN-T(Pruned)-L model is constructed, using an Encoder-Block layer configuration of 2,2,4,5,4,2, but without the inclusion of the Block3 module, GELU activation function, or CTC loss function. As shown in Table 8, the model with this configuration has a parameter count of 65.1M, and the word error rates on the Dev+ and Test+ datasets are 2.04% and 2.23%, respectively, indicating that the model's performance still requires further optimization. The preliminary results of this experiment suggest that, although the model can complete basic recognition tasks without too many additional modules, its recognition accuracy and efficiency are suboptimal, highlighting the need for the support of specific modules.

Next, the improved Zipformer-RNN-T(Pruned) model is constructed, with the Encoder-Block layer configuration adjusted to 2,2,3,5,3,2, reducing the model's parameter count to 61.1M. The model then progressively incorporates the Block3 module, GELU activation function, and CTC loss function. The experimental results show that when the model fully includes the Block3 module, GELU activation function, and CTC loss function, the word error rates on the Dev+ and Test+ datasets decrease to the lowest levels of 1.95% and 2.11%, while the real-time factor (RTF) remains stable at 0.002. This indicates that the collaboration of the Block3 module, GELU activation function, and CTC loss function significantly improves the model's recognition accuracy while maintaining high real-time performance and relatively low computational complexity, validating the importance of these components in phoneme recognition tasks.

Subsequently, when the Block3 module is removed, the word error rate on the Dev+ dataset increases to 1.99%, and there is also a slight increase in the Test+ dataset's word error rate. Similarly, when the GELU activation function or CTC loss function is removed, a slight increase in the word error rate is observed, further confirming the indispensability of these components for the model's performance. The removal of the GELU activation function leads to a decline in the model's nonlinear representation capability, affecting its generalization ability. On the other hand, removing the CTC loss function reduces the model's performance in alignment tasks, leading to an increase in the word error rate.

These results show that the introduction of each component plays a crucial role in the model's performance. The Block3 module helps the model better handle complex speech features, the GELU activation function provides stronger nonlinearity, and the CTC loss function effectively solves the alignment problem between the input and output sequences.

## 4.5 Inference experiments

In machine learning, inference is the process of predicting or classifying test data using a model that has already been trained.In this experiment, five search algorithms are used to inference the Zipformer-RNN-T(Pruned) model.

As shown in Table 9, the analysis in terms of word error rate shows that fast-beam-search has the highest word error rate, which is 50.3% and 46.74% in Dev+ and Test+ respectively,

**Table 9. Search algorithm performance comparison.**

| Model(Train+) | Dev+ | | Test+ | |
|---|---|---|---|---|
| | WER | RTF | WER | RTF |
| fast-beam-search | 50.30% | 0.005 | 46.74% | 0.006 |
| fast-beam-search-nbest | 5.79% | 0.016 | 5.88% | 0.016 |
| fast-beam-search-nbest-oracle | 2.33% | 0.007 | 2.36% | 0.007 |
| modified-beam-search | **1.95%** | 0.008 | **2.11%** | 0.008 |
| greedy-search | **1.95%** | **0.002** | **2.11%** | **0.002** |

which is due to the fact that the algorithm expands multiple candidate paths at each step, which leads to the accumulation of errors and thus increases the word error rate. In comparison, fast-beam-search-nbesth and fast-beam-search-nbest-oracle perform significantly better, but both have a real-time factor of 0.016 and 0.007, respectively, which is higher than fast-beam-search's 0.006. The WER of modified beam search and greedy search are the same, at 1.95% and 2.11% on Dev+and Test+, respectively. However, the real-time factor of greedy search is only 0.002, which is much lower than the 0.008 of modified beam search. The greedy search has the lowest real-time factor while ensuring a low WER.

## 4.6 Phonological analysis of Mandarin

The Zipformer-RNN-T(Pruned) model based on the Zipformer-RNN-T(Pruned) model shown in Fig 12 suffers from the following typical confusions in predicting Mandarin phonemes:Firstly, in terms of vowel confusion, the bilabial bursts /B/ and /P/ are often confused with each other due to subtle differences in amplitude and spectrum in the speech signal; Secondly, for front and back nasals, the difference between /IN/ and /ING/ is mainly in the presence or absence and duration of the nasal rhyme-final; Finally, in the case of the distinction between flat and warped consonants, such as /Z/ and /ZH/, /S/ and /SH/, /C/ and /CH/, the key to the distinction lies in the subtle differences in the articulatory position of the tip of the tongue. However, these subtle pronunciation differences are affected by a variety of factors such as individual differences, variation in speech rate, pronunciation habits, and dialectal environments, making them difficult to clearly define in actual recognition.

Fig 13 shows the distribution of accuracy in predicting Mandarin phonemes based on the Zipformer-RNN-T(Pruned) model. Of the 66 phonemes, 64 (97%) of the phonemes had an accuracy of greater than 95%, with only two phonemes lying between 90% and 95%, /OO/ and /UA/, with the highest accuracy being /UENG/, due to the extremely low frequency of this factor in the dataset, which contained only two samples. The model proposed in this paper performs with high accuracy on most of the phonemes, demonstrating the effectiveness and potential of the model in the task of Mandarin phoneme recognition.

## 5 Conclusion

To reduce the word error rate (WER) and model complexity in Mandarin phoneme recognition, this paper proposes an improved Mandarin phoneme recognition model based on the Zipformer-RNN-T(Pruned) architecture. This model incorporates several innovative improvements on top of the original Zipformer encoder and RNN-T decoder framework. First, structural optimizations are introduced for the Zipformer Block and Pred Network,

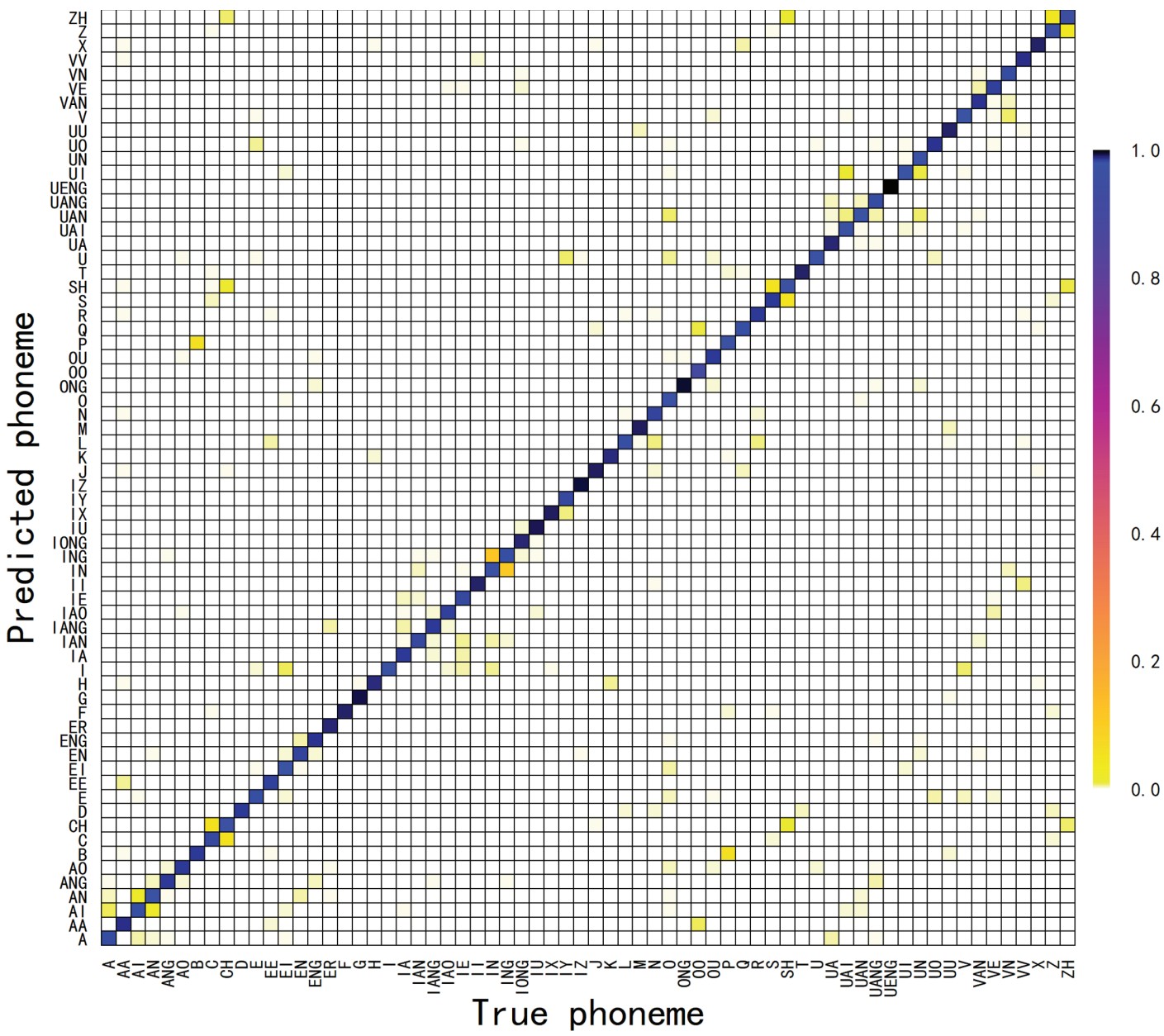

**Fig 12. Confusion matrix for Mandarin phoneme recognition based on Zipformer-RNN-T(Pruned) model.**

enabling the model to effectively reduce computational complexity while maintaining accuracy. Secondly, a hybrid Pruned RNN-T/CTC Loss strategy is proposed, which simultaneously introduces CTC loss and RNN-T loss during training to enhance the model's alignment capability and improve Mandarin phoneme recognition performance. The experimental results demonstrate:

- The improved model demonstrates outstanding performance across both benchmark datasets. On the AISHELL1-PHONEME dataset, it achieves low Word Error Rates (WER) of 1.92% on the development set and 2.12% on the test set. On the ST-CMDS-PHONEME

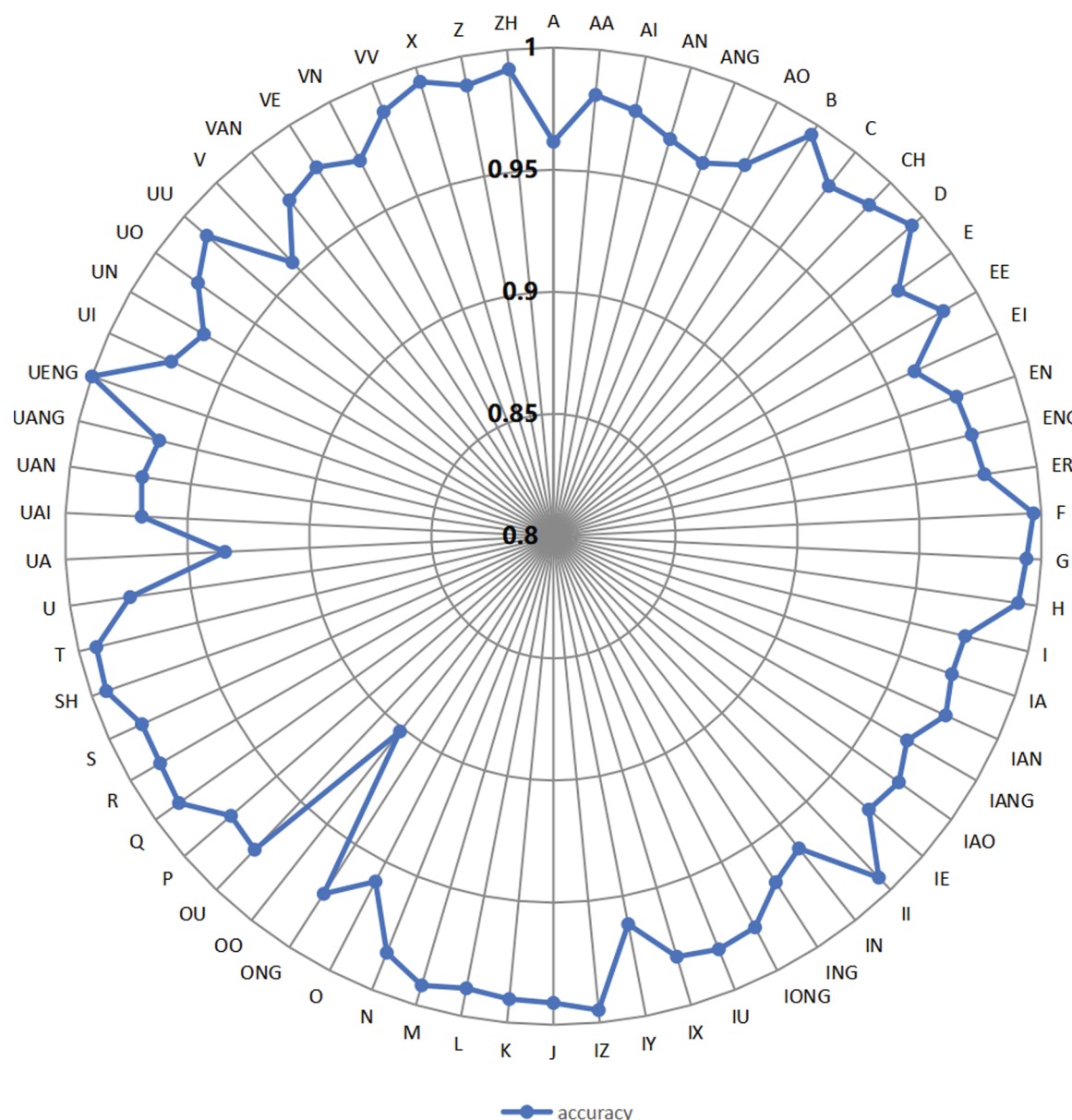

**Fig 13. Accuracy of Mandarin phonemes based on Zipformer-RNN-T(Pruned) model.**

dataset, the corresponding WERs are maintained at 4.28% and 4.51%, respectively. With only 61.1M parameters, the model holds a significant advantage over other mainstream approaches. Its efficiency and low complexity make it highly deployable in real-world scenarios, particularly on resource-constrained devices.

- To further validate the model's performance in complex environments, a simulated noise experiment was conducted. The model was tested on Train+, Dev+, and Test+ datasets containing background noise to assess its robustness. The experimental results show that, even in noisy environments, the proposed model still achieves low WER, demonstrating its strong adaptability to noise and external disturbances. The success of this experiment

proves that the improved model can maintain good performance in real-world speech recognition tasks, especially in complex noise environments.

- In addition, an ablation experiment was carried out by progressively removing or modifying components of the model to to analyse the impact of each module on model performance. The experimental results confirmed the importance of the modified modules, particularly the improved Zipformer Block and Pred Network, in enhancing the model's performance. Furthermore, the optimal value for the CTC loss weight was determined through experimentation, which further improved the model's recognition accuracy and training stability.

- To better understand the model's performance in phoneme recognition, an in-depth analysis of Mandarin phoneme misrecognition patterns was conducted. By systematically studying common issues in Mandarin phoneme recognition, this paper revealed the typical confusion patterns of phonemes and provided experimental support and theoretical foundation for future tasks such as Mandarin pronunciation evaluation. These analyses not only helped identify the weak points of the model in the Mandarin phoneme recognition task but also provided targeted solutions to improve model performance.

However, in the context of practical applications for Chinese phoneme recognition tasks, there is a relative scarcity of Chinese reading evaluation datasets, which limits the effectiveness of model training and evaluation. This issue results in larger recognition errors when handling phonemes with significant pronunciation differences. To further enhance the model's generalization ability and robustness, future research will focus on exploring data augmentation techniques and self-supervised learning methods. By incorporating more diverse training data and unsupervised learning strategies, the goal is to address the problem of data scarcity and improve the model's performance in real-world applications, particularly in adapting to unknown phonemes and complex speech environments. Future work will concentrate on how to strengthen the model's generalization ability through dataset expansion and model architecture improvements, providing more reliable and efficient solutions for Mandarin phoneme recognition applications.

## Author contributions

**Funding acquisition:** Baohua Yu.

**Methodology:** Zhaohui Du, Xiaofeng Zhao, Lin Li.

**Resources:** Lin Li, Lijiang Miao.

**Validation:** Zhaohui Du, Xiaofeng Zhao.

**Writing – original draft:** Zhaohui Du.

**Writing – review & editing:** Zhaohui Du, Baohua Yu.

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
