## [Decision Letter · Decision Letter 0]

7 Mar 2025

PONE-D-24-60568A Study on Phonemes Recognition Method for Mandarin Pronunciation Based on Improved Zipformer-RNN-T(Pruned) ModelingPLOS ONE

Dear Dr. yu,

Thank you for submitting your manuscript to PLOS ONE. After careful consideration, we feel that it has merit but does not fully meet PLOS ONE’s publication criteria as it currently stands. Therefore, we invite you to submit a revised version of the manuscript that addresses the points raised during the review process.

Please revise the paper by considering the reviewer's comments.

We look forward to receiving your revised manuscript.

Kind regards,

Jie Zhang

Academic Editor

PLOS ONE

Journal Requirements:

3. Thank you for stating the following financial disclosure: This research was funded by [the Bing-tuan Science and Technology Public Relations Project “A Data-driven Regional Smart Education Service Key Technology Research and Application Demonstration”] grantnumber [2021AB023]., 

Reviewers' comments:

Reviewer's Responses to Questions

**Comments to the Author**

1. Is the manuscript technically sound, and do the data support the conclusions?

Reviewer #1: Yes

2. Has the statistical analysis been performed appropriately and rigorously? 

Reviewer #1: Yes

3. Have the authors made all data underlying the findings in their manuscript fully available?

Reviewer #1: Yes

4. Is the manuscript presented in an intelligible fashion and written in standard English?

Reviewer #1: Yes

5. Review Comments to the Author

Reviewer #1: The proposed model demonstrates significant advancements in recognition accuracy and real-time processing capabilities, achieving WERs of 1.92% on the Dev set and 2.12% on the Test set with a real-time factor (RTF) of 0.002. These results highlight the model's practical utility for real-time applications, such as voice assistants and live translation systems. The use of 61.1M parameters ensures a lightweight architecture that balances accuracy with computational efficiency, making it highly suitable for deployment in resource-constrained environments. Furthermore, the model’s robustness in noisy environments is emphasized through experiments using the MUSAN dataset, where it maintains low WERs even under challenging acoustic conditions. This adaptability to diverse environments enhances its applicability in real-world scenarios with prevalent noise interference.

However, the paper exhibits several limitations. The introduction lacks clarity and a logical flow, particularly in justifying the necessity of the proposed model. Although challenges in phoneme recognition are discussed, the connection between these challenges and the choice of the Zipformer variant is insufficiently articulated. Additionally, the importance of RTF optimization for real-time Chinese phoneme recognition is not adequately supported with concrete examples or detailed reasoning. Another major drawback is the reliance on the AISHELL1-PHONEME dataset, which limits the generalizability of the model. Evaluating performance solely on a single dataset raises concerns about its effectiveness in different linguistic contexts, such as conversational or regional Mandarin. Furthermore, the paper does not sufficiently emphasize the model's relevance to Chinese phoneme recognition, as specific challenges like tonal variations and real-time processing requirements are not clearly tied to the proposed approach.

To improve the study, incorporating additional datasets such as L2-ARCTIC or augmenting AISHELL1-PHONEME with multilingual or augmented data would strengthen the model’s validation. If other datasets are unavailable, data augmentation techniques could be explored to simulate diverse linguistic contexts. Additionally, the introduction should more effectively highlight the specific challenges in Chinese phoneme recognition and how the Zipformer variant addresses these issues, supported by clear examples and reasoning. Despite these limitations, the model presents promising results in accuracy, efficiency, and noise robustness, making it a strong candidate for real-world applications. Enhancing the scope of evaluation and refining the introduction would further solidify the paper’s contributions to Mandarin phoneme recognition research.

6. PLOS authors have the option to publish the peer review history of their article (what does this mean?). If published, this will include your full peer review and any attached files.

Reviewer #1: No

---

## [Author Response · Author response to Decision Letter 1]

16 Apr 2025

Dear Editor,

Thank you for allowing us to resubmit the manuscript and for the opportunity to address the reviewers' comments. We have carefully revised the manuscript based on the feedback provided in the attached summary of reviewers' comments (as shared by your office).

We are now uploading the following documents:

1.A point-by-point response to the reviewers' comments (under "Response to Reviewers"), addressing each concern raised in the attached summary.

2.A highlighted version of the revised manuscript (under "Highlighted PDF"), with all changes marked in yellow.

3.A clean version of the revised manuscript (under "Main Manuscript").

Please let us know if any further adjustments are needed. We appreciate your time and consideration.

Best regards,

Zhaohui Du et al

---

## [Editor Report · Decision Letter 1]

20 Apr 2025

A Study on Phonemes Recognition Method for Mandarin Pronunciation Based on Improved Zipformer-RNN-T(Pruned) Modeling

PONE-D-24-60568R1

Dear Dr. yu,

We’re pleased to inform you that your manuscript has been judged scientifically suitable for publication and will be formally accepted for publication once it meets all outstanding technical requirements.

Kind regards,

Jie Zhang

Academic Editor

PLOS ONE

Additional Editor Comments (optional):

The authors have adequately addressed the reviewer's comments.
---

## [Editor Report · Acceptance letter]

PONE-D-24-60568R1

PLOS ONE

Dear Dr. Yu,

I'm pleased to inform you that your manuscript has been deemed suitable for publication in PLOS ONE. Congratulations! Your manuscript is now being handed over to our production team.

Kind regards,

on behalf of

Dr. Jie Zhang

Academic Editor

PLOS ONE